



# The portable ice nucleation experiment PINE: a new online instrument for laboratory studies and automated long-term field observations of ice-nucleating particles

Ottmar Möhler[1], Michael Adams[2,*], Larissa Lacher[1,*], Franziska Vogel[1,*], Jens Nadolny[1], Romy Ullrich[1], Cristian Boffo[3,4], Tatjana Pfeuffer[3], Achim Hobl[3], Maximilian Weiß[5], Hemanth S. K. Vepuri[6], Naruki Hiranuma[6], and Benjamin J. Murray[2]

[1]Institute of Meteorology and Climate Research, Karlsruhe Institute of Technology, Karlsruhe, Germany
[2]School of Earth and Environment, University of Leeds, Leeds, UK
[3]Bilfinger Noell GmbH, Würzburg, Germany
[4]Fermi National Accelerator Laboratory, IL, USA.
[5]Palas GmbH, Karlsruhe, Germany
[6]Department of Life, Earth and Environmental Sciences, West Texas A&M University, TX, USA
[*]These authors contributed equally to this work.

**Correspondence:** Ottmar Möhler (ottmar.moehler@kit.edu)

**Abstract.** Atmospheric ice-nucleating particles (INP) play an important role in determining the phase of clouds, which affects their albedo and lifetime. A lack of data on the spatial and temporal variation of INPs around the globe limits our predictive capacity and understanding of clouds containing ice. Automated instrumentation that can robustly measure INP concentrations across the full range of tropospheric temperatures is needed in order to address this knowledge gap. In this study, we demon-

strate the functionality and capacity of the new Portable Ice Nucleation Experiment (PINE) to study ice nucleation processes and to measure INP concentrations under conditions pertinent for mixed-phase clouds, with temperatures from about $-10\,°C$ to about $-38\,°C$. PINE is a cloud expansion chamber which avoids frost formation on the cold walls, and thereby omits frost fragmentation and related background ice signals during the operation. The development, working principle, and treatment of data for the PINE instrument is discussed in detail. We present extensive laboratory based tests where PINE measurements

were compared with those from the established AIDA (Aerosol Interaction and Dynamics in the Atmosphere) cloud chamber. The results show good agreement of PINE with AIDA for homogeneous freezing of pure water droplets and the immersion freezing activity of mineral aerosols. Results from a first field campaign conducted at the Atmospheric Radiation Measurement (ARM) Southern Great Plains (SGP) observatory in Oklahoma, USA, from October 1 to November 14, 2019 with the latest PINE design (a commercially available PINE chamber) are also shown, demonstrating PINE's ability to make automated

field measurements of INP concentrations at high time resolution of about 8 minutes with continuous wall temperature scans between $-5$ and $-35\,°C$. During this field campaign, PINE was continuously operated for 45 days in a fully automated and semi-autonomous way, demonstrating the capability of this new instrument to be also used for longer term field measurements and INP monitoring activities in observatories.



## 1 Introduction

Atmospheric ice-nucleating particles (INP) induce ice formation in atmospheric clouds, and by that are important for initiating precipitation in mixed-phase clouds and determining the phase of clouds, their albedo, lifetime and other important properties (DeMott et al., 2010). However, the details of these aerosol-cloud-climate interactions remains highly uncertain (Boucher et al., 2013; Fan et al., 2017; Lohmann, 2017). This is partly due to the fact that such clouds are rather complex systems, and that the

knowledge on the formation, the concentration and the fate of ice crystals is still uncertain (Heymsfield et al., 2017; Korolev et al., 2017).

In the absence of homogeneous freezing, the cloud ice phase is initiated in various ways by a very small fraction of atmospheric aerosol particles (INPs) (Vali et al., 2015). In mixed-phase clouds, immersion freezing is thought to be the dominating freezing mechanism (de Boer et al., 2011; Hande and Hoose, 2017; Hoose et al., 2010). Vergara-Temprado et al. (2018) showed

INPs to have a strong control of cloud reflectivity over the Southern Ocean. Mülmenstädt et al. (2015) and Field and Heymsfield (2015) showed the ice or snow phase to exist in a large fraction of precipitating clouds, in particular over the continents. This underlines the importance of INPs for cloud radiative properties and precipitation formation, but it should be noted here that the cloud ice phase not only depends on the primary ice formation by INPs, but is also largely influenced by a cascade of secondary ice formation and interaction processes, in particular at temperatures above $-15\,°C$ (Field et al., 2016). Increased ice

crystal concentrations can e.g. lead to rapid cloud glaciation and associated dissipation (Campbell and Shiobara, 2008; Paukert and Hoose, 2014), as also observed recently in a laboratory cloud chamber experiment (Desai et al., 2019) .

At higher altitudes with temperatures below about $-35\,°C$, cirrus cloud ice crystals can either be formed by homogeneous freezing of aqueous aerosol particles at relatively high ice supersaturations (Koop et al., 2000; Kärcher and Lohmann, 2002), or by heterogeneous ice nucleation processes at lower ice supersaturations (Hoose and Möhler, 2012; Kärcher and Lohmann,

2003; Krämer et al., 2016; Murray et al., 2010). As in the mixed-phase cloud regimes, the heterogeneous pathways of cirrus ice crystal formation are limited and controlled by the abundance of INPs in the upper troposphere, in addition to other factors like dynamic, thermodynamic or kinetic processes (Heymsfield et al., 2017).

Throughout the troposphere, INPs are difficult to identify and to quantify due to their low and largely variable number fraction to the total aerosol concentration (DeMott et al., 2010; Kanji et al., 2017). This fraction strongly depends not only

on temperature and relative humidity conditions, but also on the particle type, size, and surface properties (Pruppacher and Klett, 2010; Holden et al., 2019). Nevertheless, cloud, weather and climate models need to formulate and quantify primary ice formation as accurately as possible (Vergara-Temprado et al., 2018; Waliser et al., 2009). This is achieved by calculating the abundance of INPs with parameterizations based on either laboratory ice-nucleation experiments (Hoose and Möhler, 2012; Murray et al., 2012; Sesartic et al., 2013; Spracklen and Heald, 2014; Vergara-Temprado et al., 2018) or field measurements

(DeMott et al., 2010; McCluskey et al., 2018; Tobo et al., 2013; Wilson et al., 2015). A number of different parameterizations for the various pathways of atmospheric ice nucleation in supercooled liquid and cirrus clouds have been developed under



different assumptions, based on either temperature and time dependent ice formation rates according to classical nucleation rate formulations (Barahona and Nenes, 2009; Kärcher and Lohmann, 2002, 2003), the number concentration of larger aerosol particles (DeMott et al., 2010, 2015), or the temperature-dependent ice nucleation active site (INAS) density on the surface of aerosol particles (Connolly et al., 2009; Harrison et al., 2019; Niemand et al., 2012; Ullrich et al., 2017).

The proper use of aerosol particle specific INP parameterizations, however, requires aerosol type specific knowledge of parameters like number concentration and size distribution, needed as input to the calculation and prediction of INP concentrations. The application of these ice nucleation parameterizations can be challenging, because of limitations in aerosol characterization in field campaigns and modelling studies. In particular, information on the types, chemical nature, and mixing state of aerosol particles is often missing, but may have a strong impact on the ice nucleation activity or INP abundance (Möhler et al., 2008). At present, the atmospheric INP data that we can compare with global fields of INP concentrations are extremely limited in spatial, temporal and concentration ranges (Burrows et al., 2013; Vergara-Temprado et al., 2017). Hence, there is an urgent need for more INP observation and monitoring, not only for constraining INP predictions by models and representing a fuller range of INP sources in those models, but also to extend the data base for a better understanding of temperature dependent INP concentrations throughout the atmosphere and the year.

Existing measurements of ambient INP concentrations at mixed-phase cloud temperatures (Kanji et al., 2017) show a great variability not only across the temperature range from about $-5\,°C$ to $-35\,°C$ (10 orders of magnitudes), but also at a single temperature ($\sim 4$ orders of magnitude). Different aerosol types were found to dominate the INP population at specific temperatures. While high temperature INPs are typically associated with biological particles (e.g., DeMott et al., 2010; Creamean et al., 2013; Prenni et al., 2013; Mason et al., 2015b; DeMott et al., 2015; O'Sullivan et al., 2018), their atmospheric implication remains uncertain (Després et al., 2012; Hummel et al., 2018). Marine aerosol particles were identified to be ice active at $T > -30\,°C$ (Alpert et al., 2011; Brier and Kline, 1959; DeMott et al., 2015; Mason et al., 2015a, b; McCluskey et al., 2018; Wilson et al., 2015). They might be an important source for INPs in the absence of more ice active aerosol particles (Burrows et al., 2013; Vergara-Temprado et al., 2017; Wilson et al., 2015). Mineral dust particles are very efficient INPs at $T < -20\,°C$ (Boose et al., 2016c; Harrison et al., 2019; Ullrich et al., 2017) and may dominate the INP number concentrations in many locations (Atkinson et al., 2013; Sanchez-Marroquin et al., 2020; Tobo et al., 2019).

Most of these measurements were only sensitive for immersion freezing INPs in the temperature range of mixed-phase clouds, and were carried out at boundary layer field sites which were considered to be predominantly influenced by different aerosol types. Measurements in the free troposphere were either performed at high altitude mountain stations (Boose et al., 2016a, b; DeMott et al., 2003a; Conen et al., 2015; Lacher et al., 2018a, b) or with aircraft-based measurements (Rogers et al., 2001; DeMott et al., 2003b; Prenni et al., 2009; Pratt et al., 2010; Eidhammer et al., 2010; Field et al., 2012), but most of them were also limited to measure immersion freezing INPs at higher temperatures. DeMott et al. (2003b) also measured the concentration of INPs active in the deposition mode at temperatures below $-40\,°C$.

The identification of INP types in ambient air remains challenging. Most ambient studies focus on sampling INPs in campaigns over a limited time period and focused on specific air masses like Saharan dust events (Boose et al., 2016b), biogenic source regions (O'Sullivan et al., 2018) or marine environments (Mason et al., 2015a), or use back trajectories to identify



source regions (e.g., Lacher et al., 2017; Wex et al., 2019). Such approaches are not only in need of high-time resolution INP measurements to characterize changing air masses, but also long-term monitoring of INPs to capture the bigger picture and not only short-term periods of the atmosphere.

An increasing number of new methods and instruments for INP measurements have been developed and compared to each other during the previous years (DeMott et al., 2011; Hiranuma et al., 2015; Wex et al., 2015; DeMott et al., 2018). The most recent and comprehensive INP instrument and method intercomparison study was the Fifth International Workshop on Ice Nucleation Research (FIN-2), and many of the latest developments for atmospheric INP measurements are included and described with respective references in the overview paper by DeMott et al. (2018). Most of the INP methods showed reasonable

agreement with each other, but most of them are time and operator intensive. A general feature is that available online instruments can only be operated periodically, and offline methods base on aerosol samples have poor time resolution depending on required aerosol sampling time of hours to days. All existing methods require intensive man-power and time for operation or offline analysis. The low time resolution of offline techniques challenges the comparison to potential driving factors for ice nucleation, as e.g. the size and chemistry of the aerosol population. For such measurements, online INP instruments are

desirable, having a high-time resolution of minutes.

    This paper presents the development, technical description, working principle, as well as first laboratory and field applications of the new Portable Ice Nucleation Experiment PINE. PINE is the first fully automated instrument for laboratory ice nucleation studies and long-term field observations of INPs in a wide temperature range from $-10\,°C$ to about $-60\,°C$, including mixed-phase cloud and cirrus cloud regimes and related primary ice formation processes. Similar to the AIDA (Aerosol

Interaction and Dynamics in the Atmosphere) cloud simulation chamber, PINE is based on a pumped expansion principle to induce ice and water supersaturated conditions for aerosol particles sampled either from laboratory setups or natural environments. The instrument is operated in repeated cycles of sampling the aerosol into a pre-cooled cloud chamber, activating the aerosol particles as supercooled droplets and ice crystals by expanding the air inside the cloud chamber, and refilling the cloud chamber with fresh aerosol for the next cycle (see section 4).

**2   Basic principles and milestones of the PINE development**

The idea for PINE resulted from almost 20 years of experience operating the AIDA facility for cloud experiments at simulated conditions of up-drafting atmospheric air parcels. Cloud formation in the rigid but large AIDA chamber with a volume of $84\,m^3$ is induced in a controlled way by lowering the pressure at different rates, starting from well controlled thermodynamic conditions (Möhler et al., 2003, 2005). With a volume of only about $10\,L$, the PINE cloud simulation chamber is much smaller,

transportable, and operated in a fully automated sequence. Similar to the AIDA cloud chamber, PINE also uses the principle of pressure reduction by controlled pumping of air out of the cloud chamber. By that, the temperature in the chamber decreases due to expansion cooling, while the relative humidity increases. This causes the aerosol particles, which are present in the chamber prior to the expansion, to act as Cloud Condensation Nuclei (CCN) and/or INPs to form liquid cloud droplets and ice crystals, depending on the temperature, ice supersaturation and the type of aerosol. The starting temperature of each expansion





run, and thereby the temperature range of ice formation and INP detection, can be set in a wide range from about $-10\,°C$ to $-60\,°C$, depending on the capacity of the cooling system. Large aerosol particles, droplets and ice crystals are measured and counted with an optical particle counter (OPC) mounted directly to the vertically oriented pump tube below the cloud chamber. PINE can be operated both for ice nucleation research in the laboratory, and for INP measurements in field campaigns or long term monitoring activities.

The first version of PINE was successfully tested in January 2016. It consisted of a simple perplex chamber of $10\,L$ volume with manual valve and flow control, and a welas 2300 single particle optical detector from Palas GmbH, Karlsruhe, Germany. This setup was operated in a cold room at temperatures around $-10\,°C$ and sampled Snomax© aerosol particles for first proof-of-concept studies of immersion freezing in the small cloud expansion chamber. The critical development idea for PINE was to pass the total pump flow during the cloud expansion cycle through the optical particle counter directly attached to the pump

line (see patent applications DE 10 2018 210 643 A1 and US2020/0003671 A1). Another prototype chamber of about $7\,L$ volume was then built of stainless steel and also operated in a cold room for further proof-of-concept experiments.

Based on the development and operational experience with the prototype versions, we developed the first mobile versions PINE-1A and PINE-1B with their own cooling systems and a control system for semi-autonomous operation during laboratory ice nucleation measurements and field INP observations. Because both systems are almost identical, we only refer to PINE-1A

in the following sections, for simplicity. PINE-1A can be operated in a temperature range from $-10\,°C$ to about $-40\,°C$, was characterized in a series of laboratory experiments, and was used in a first field campaign (Adams et al., in prep.). As a next step, the version PINE-c was developed, which is now commercially available from Bilfinger Noell GmbH in Germany (see https://www.noell.bilfinger.com/pine/#c167514). PINE-c is operated in the same way as PINE-1A, but received a few new components and features making it more compact and autonomous for operation in both field and lab studies. This will be

detailed in Sect. 3, together with a general technical description of the new PINE instrument. The typical working principle and operation of PINE will be explained in Sect. 4. In Sect. 5 we summarize and discuss some first results from laboratory test and characterization experiments of PINE-1A in comparison with the AIDA cloud chamber. Finally, in Sect. 6, we will present and discuss some results from a first field application of PINE-c, which continuously measured during all 45 days of a field campaign at the DOE SGP site in Oklahoma from October 1st to November 14th, 2019.

## 3 PINE instrument setup

As illustrated in Fig. 1, PINE consists of 5 major parts, (I) an inlet system, (II) a cloud chamber, (III) a cooling system, (IV) a particle detection system, and (V) a control and data acquisition system. Figure 2 shows a simplified schematics of the PINE setup in the different operational flow configurations that will be discussed in Sect. 3.

The inlet system (Fig. 1, part I) is composed of an inlet or sampling tube, a diffusion dryer, a humidity sensor and a bypass

flow section with aerosol particle filter for background measurements. The relative humidity, measured with a dew point sensor (Vaisala DRYCAP® DMT143) has to be high enough to allow cloud droplet formation upon expansion cooling, and at the same time low enough to avoid frost formation on the chamber walls (see Sects. 4 and 5). Both the prototype version PINE-1A and





the commercial version PINE-c (see Tab.1 and Sect.6), are equipped with two nafion membrane diffusion dryers (Permapure, MD-700-24S-1, length 62 cm) in parallel, in order to reduce the flow through one single dryer and by that enhance the drying
efficiency.

Figure A1 shows the PINE sample flow dryer setup with two nafion diffusion dryers and other major components. The sample flow passes the straight nafion tube of $1.7\,\mathrm{cm}$ diameter and $62\,\mathrm{cm}$ length from top to bottom. The nafion tube is located inside an airtight stainless steel tube of $2.5\,\mathrm{cm}$ diameter. A second air flow is passing the annular gap between the coaxial nafion and stainless steel tubes from bottom to top (counter flow arrangement). For simplicity, the PINE system uses ambient
air for this counter flow, but at reduced absolute pressure. The absolute pressure reduction also reduces the water vapour partial pressure compared the sample flow inside the nafion tube at ambient pressure. This water vapour partial pressure difference across the nafion membrane, which is permeable for water molecules, drives a diffusional flow of water molecules from the sample flow to the counter flow. The molar flux of water molecules increases with the pressure difference across the membrane and the residence time of the sample air inside the nafion tube. As seen in Fig. A2, the drying efficiency increases with pressure
reduction. The pressure of the counter flow air is controlled with a pressure controller (Wagner-MSR type P-702), located between the dryer and the vacuum pump, and the volumetric flow rate of the counter flow air is controlled with a critical orifice at the inlet side. In comparison to conventionally used diffusion dryers with water adsorption material, the membrane dryers used in the PINE setup have the great advantage that they can be operated for long-term without decreasing their drying efficiency.

Because the current PINE instrument versions are typically operated with a sample flow rate of up to $4\,\mathrm{L\,min^{-1}}$ (see Sect. 4), two nafion dryers are used in parallel operation, in order to limit the sample flow through each dryer to $2\,\mathrm{L\,min^{-1}}$. If needed, the dryers can then be operated with a maximum pressure difference of about $800\,\mathrm{hPa}$ to achieve a high drying efficiency with a drop in the sample flow dew point temperature of at least $10\,°\mathrm{C}$. As mentioned above, the frost point temperature of the sample air should be close to the wall temperature of the PINE cloud chamber. If the sampled air is too humid, frost may form
at the coldest wall sections, potentially causing and an increasing ice background due to frost artefacts. However, this was not the case when operating PINE-1A during a first field application (Adams et al., in prep.) for several weeks at temperatures below $-25\,°\mathrm{C}$ and sample air frost point temperatures around $-15\,°\mathrm{C}$. This was tested by passing the sample flow through the particle filter (see Fig. 2) resulting in zero particle counts after about 5 consecutive runs (Fig. A3; see also Adams et al., in prep.). This means that when the sample air is passing through the bypass particle filter, the system detects neither aerosol
particles, nor activated droplets nor ice crystals.

The heart of a PINE instrument is a temperature controlled cloud chamber (Part II in Fig. 1). The PINE-1A cloud chamber has a volume of about $7\,\mathrm{L}$ and is made of stainless steel, with a central cylindrical part and two cones at the top and the bottom (Tab. 1). With the cooling system (part III in Fig. 1), the wall temperature of the cloud chamber can either be precisely controlled at a constant value, or changed at constant cooling or heating rates of up to $0.3\,°\mathrm{C\,min^{-1}}$. The PINE-1A cloud chamber is
temperature-controlled with an ethanol bath chiller (Lauda RP 855; Lauda-Königshofen, Germany). The wall temperature of the chamber is measured with three thermocouples attached to the outer chamber walls at three different locations. The gas temperature inside the cloud chamber is also measured with three thermocouples located in the bottom, middle and upper



section of the chamber about 5 cm off the wall (see Fig. A4). All thermocouples have been calibrated to a reference sensor (Lake Shore, Model PT-103, Sensor Typ Platinum Resistor) with an accuracy of $\pm 0.1\,°\mathrm{C}$. A minimum wall temperature of about

$-33\,°\mathrm{C}$ can be reached with PINE-1A. With additional expansion cooling of the chamber volume (see Sect. 4), a minimum gas temperature of about $-33\,°\mathrm{C}$ is then reached for ice activation of the aerosol particles.

PINE-c is equipped with a thin-walled aluminium cloud chamber with a slightly larger volume of $10\,\mathrm{L}$ as compared to PINE-1A (see Table 1). Mainly for thermal insulation, the cloud chamber is located in an evacuated stainless steel container and is cooled with a Stirling cooler (Thales, LPT9310). The combination of the low mass and heat capacity of the thin-walled

cloud chamber and the high cooling power of the stirling cooler allows to cool the PINE-c cloud chamber at a rate of up to approximately $0.6\,°\mathrm{C}\,\mathrm{min}^{-1}$ without any notable effects of measurement disturbance (see Sect. 6). The heating rate of the chamber can also automatically be set to a value up to $0.6\,°\mathrm{C}\,\mathrm{min}^{-1}$. By that, faster temperature scans than with PINE-1A can be achieved for temperature-dependent ice nucleation and INP measurements. PINE-c can also be cooled to a lower wall temperature of -60°C and can therefore be operated for ice nucleation experiments and INP measurements at cirrus cloud

temperatures.

The PINE particle detection system (part IV in Fig. 1) consists of an OPC connected to the control and data acquisition system (part V in Fig. 1). Depending on the OPC type, aerosol particles, activated cloud droplets and ice crystals are detected during the different run modes as described in Sect. 4. The OPC is mounted to the pump tube, with a minimum distance to the cloud chamber in order to minimize warming of the cold air flow from the cloud chamber and by that avoid evaporation

of supercooled cloud droplets and sublimation or melting of ice crystals. PINE-1A is equipped with a welas 2500 sensor connected to a Promo© 2000 system (Palas GmbH, Karlsruhe, Germany). The same sensor has been operated for many years at the AIDA cloud chamber for cloud droplet and ice crystal detection (Möhler et al., 2006; Wagner and Möhler, 2013). It measures aerosol particles, water droplets and ice crystals with optical sizes between $0.7$ and $220\,\mu\mathrm{m}$ diameter, depending on the sensitivity setting of the photomultiplier measuring the intensity of light scattered by single particles when passing the

optical detection volume (ODV). The welas sensor records for each detected particle the time of detection, the transit time through the ODV, and the intensity of light scattered into a range of scattering angles around $90°$ (sideward scattering). This optical arrangement is favourable for the selective detection of a-spherical ice crystals, which are measured at a larger optical size than spherical droplets of the same volume and can therefore more easily be distinguished from droplets by setting a simple threshold for the optical size (see Sect. 4).

The welas 2500 sensor has a well confined ODV with a sample flow cross section area $A_w = 0.24\,\mathrm{mm}^2$ and a length $l_w = 0.31\,\mathrm{mm}$. Because the transect time $\tau_w$ of particles through the ODV is also measured, the sample flow rate through the ODV can be calculated as

$$F_w = \frac{A_w l_w}{\tau_w}. \tag{1}$$

With the count rate $c_p$ of detected particles, one can then calculate the particle number concentration

$$n_p = \frac{c_p}{F_w}. \tag{2}$$





On average, the ratio of the volume flow through the ODV to the total volume flow through the welas 2500 sensor is about 0.105. This means that the sensor detects only about $10\,\%$ of the particles sampled from the cloud chamber. The PINE-c version uses a new OPC called fidas-pine (Palas GmbH, Karlsruhe, Germany). This new OPC was developed especially for the PINE-c instrument and analyses the full sample flow of up to $5\,\mathrm{L\,min^{-1}}$ for particles in a size range similar to the welas 2500 sensor.

For PINE-c, the particle number concentration can still be calculated with Eq.2, just by replacing the flow rate through the ODV of the welas 2500 sensor by the total sample flow rate $F_{em}$ during the expanion mode (see Sect. 4). Therefore, fidas-pine has a 10 times higher detection rate of particles, and by that a 10 times lower INP concentration detection threshold than PINE-1A.

PINE is controlled via a bespoke LabVIEW program, which sets the respective measurement condition, displays the parameters such as particle size, temperature, pressure, and flows, and saves the data internally. Metadata describing the experiment

are saved automatically using LabVIEW, such as date and time, type of OPC used and its configuration, temperature and pressure conditions.

## 4   PINE operating principle

The PINE instrument can either be used in an individual operator controlled mode for laboratory ice nucleation experiments and measurements, or in a fully automated mode for long-term field measurements and observations of INPs. The instrument's

settings during a laboratory or field campaign and the data storage systems of PINE are organized in a well-defined sequence of operations and runs. All data and metadata are saved automatically using the LabVIEW program.

An operation is defined as a specific series of runs, which can be, for example, performed at one temperature, and during a specific sampling time. Each run is composed of a sequence of three modes called "flush", "expansion", and "refill". The flow settings of PINE in these three run modes are illustrated in Fig. 2. In the flush mode (Fig. 2a), the sample flow is passed through

the cloud chamber to fill it with the aerosol under investigation. This can either be ambient air at a field station where PINE measures INP concentrations, or an aerosol generated in a laboratory for ice nucleation studies. For PINE-1A and PINE-c, the sample flow rate is limited to about $4\,\mathrm{L\,min^{-1}}$ (see Sect. 4). In the flush mode, the sample flow can also be passed through a aerosol filter for background, particle-free measurements.

In the expansion mode (Fig. 2b), the sample flow is kept constant, but switched to a bypass line around the cloud chamber.

At the same time, a valve at the chamber inlet is closed and the OPC flow rate is set to a value between 2 and $5\,\mathrm{L\,min^{-1}}$, such that the pressure in the cloud chamber is lowered at a constant rate and to a pre-defined minimum pressure. This forced gas expansion in the cloud chamber causes an adiabatic cooling and thereby an increase of the relative humidity. When the relative humidity exceeds ice or water saturation, the aerosol particles in the cloud chamber, which were sampled during the flush mode, are then activated to form ice crystals and/or liquid cloud droplets, depending on the temperature and the type of

aerosols. The number of ice crystals is measured with the OPC downstream of the chamber, and equals the number of INPs in the same sampling volume. The expansion mode flow rate $F_{em}$ is limited to 2 and $5\,\mathrm{L\,min^{-1}}$, because both the welas 2500 and fidas-pine OPCs can only be operated to a maximum sample flow rate of $5\,\mathrm{L\,min^{-1}}$. Smaller flow rates can cause cloud droplet evaporation or ice crystal sublimation in the tube connection between the cloud chamber and the OPC. The end pressure is





typically 200 to 300 hPa lower than the start pressure that is given by the pressure of the aerosol sampled during the flush
mode.

The refill mode (Fig. 2c) is the final run mode and is carried out to safely re-pressurize the PINE chamber to the start
pressure. Once this pressure is reached, the sample flow is immediately switched back to pass the cloud chamber, starting the
next run with the same series of flush, expansion and refill modes. A full run takes about 4 to 6 minutes, depending on the flush
time, the pump flow rate during the expansion mode and the end pressure. The higher the sample flow rate, the faster the air
in the cloud chamber is replenished and renewed for the next run, and the shorter the flush time can be chosen. The lower the
minimum pressure during expansion, the longer the refill mode time.

Figures 3 to 6 show results from a PINE-1A operation on March 25, 2018 during the HyICE field campaign, which includes
a series of 59 identical runs. Each run took about 6 minutes, so the whole operation lasted almost 6 hours. During this time, the
ambient total aerosol concentration varied between about 900 and 2300 cm$^{-3}$, and PINE-1A sampled ambient air at a flow rate
of 3 L min$^{-1}$. The flush time was set to 4 minutes. Each expansion was started at a wall temperature of about $-26\,°C$ with pump
flow rate of 4 L min$^{-1}$, and took about 40 seconds. An example of these 59 runs is depicted in Fig. 3, which shows the end of
the flush mode, the expansion mode and the refill mode. The data time series are plotted as a function of the time in seconds
relative to the start of the expansion mode. A temperature and pressure decrease of about 6 °C and 300 hPa, respectively, is
observed during the expansion (Fig. 3a). Here, only the data from the lowest of the three gas temperature sensors (see Fig. A4)
is plotted, which reaches a minimum value of about $-32\,°C$ at the end of the expansion after about 40 seconds.

The relative humidity is not directly measured in the PINE cloud chamber, but can be calculated from the change of the
temperature dependent water saturation pressure, assuming ice saturated conditions at the start of the expansion and omitting
water vapour sources and sinks during the expansion. As mentioned above, we assumed ice saturated conditions so that the
water vapor partial pressure at the start of expansion equals the ice saturation pressure calculated as function of the wall
temperature at start of expansion ($p_{w,0} = p_{sat,ice}(T_{g,0})$), and the corresponding saturation ration with respect to liquid water is
$S_w = 0.79$ at the same start temperature $T_{g,0} = -26\,°C$. During the expansion mode, the liquid water saturation ratio was then
calculated as

$$S_w = \frac{p_w}{p_{sat,liq}(T_g)} \tag{3}$$

with

$$p_w = p_{w,0}\frac{p}{p_0} \tag{4}$$

where $p_0$ is the pressure at start of expansion and $p$ the pressure during the expansion. It can be seen that after about 10
seconds, the so calculated $S_w$ exceeds 1 (Fig. 3b). Note that S will in reality be limited by the growth of cloud droplets,
but that conditions of $S > 1$ indicate conditions where a liquid cloud could form. This roughly corresponds with the start of
cloud droplet activation as shown in panel (c) of Fig. 3, shown by the sudden occurrence of a large number of particles with
diameters up to 10 μm. This panel shows each single particle detected by the OPC plotted as a single blue dot at the time of
occurrence and with its measured optical diameter. With ongoing pressure reduction and related cooling, a small number of





particles is detected at larger optical size, with diameters larger as the dense "cloud" of liquid droplets (Fig. 3c). Those particles are identified as ice crystals formed by immersion freezing of only a minor droplet fraction. The expansion mode stops after about 40 seconds and the chamber is refilled to ambient pressure within about 1 minute. The next run is started with the flush

mode, filling the cloud chamber again with ambient aerosol particles for the next expansion run. Refilling causes compression of the chamber air and related warming (see upper panel of Fig. 3). This also leads to the evaporation of the droplets and ice crystals after some time, however, the abrupt stop of particle recording is related to the fact that the pump flow rate through the OPC is stopped at the end of expansion, so that only a few particles are moving through the OPC detection volume during the refill mode.

For the same PINE-1A operation during the HyICE field campaign, there is little run-to-run variation for the total OPC counts per second of run time (Fig. 4). This means that PINE is able to reproduce aerosol CCN activation and super-cooled droplet formation in repeated runs at constant sampling and operation conditions, which provides a good basis for conducting series of immersion mode INP and freezing measurements at high precision. The small grey dots in this figure show the OPC count rates of individual runs, the bigger black circle the mean over all 59 runs of this operation. The variation can partly be explained by

the natural variability of the ambient aerosol concentration which also causes a variation of the droplet number concentration. As mentioned above, the aerosol number concentration varied by about a factor of two between $900$ and $2300\,\mathrm{cm^{-3}}$.

Not only cloud droplets but also ice crystals were detected during the same operation as shown by the occurrence of particles larger than $\sim 10\,\mu\mathrm{m}$ (Fig. 3c). The whole size distribution of both cloud droplet and ice crystals (Fig. 5) reveals that there is only little variation from run to run, at least for the droplet mode with maximum diameters of $\sim 10\,\mu\mathrm{m}$. Larger particles are

identified as ice crystals, and can be distinguished from the droplets by setting a size threshold close to the end (the "right edge") of the sharp droplet mode.

Based on Eq. 1, the immersion mode INP number concentration measured in one run of the PINE-1A system can then be calculated by dividing the total number $\Delta N_{ice}$ of ice crystals detected by the total volume $\Delta V_w$ of air passing the ODV of the welas OPC during the expansion mode after the formation of the supercooled liquid cloud

$$n_{INP,w} = \frac{\Delta N_{ice}}{\Delta V_w} = \frac{\Delta N_{ice}}{F_w \Delta t_{em}} \tag{5}$$

where $F_w$ is the volumetric flow rate through the optical detection volume of the welas sensor, and $\Delta t_{em}$ the duration of the expansion mode from the start of liquid cloud formation (see also Sect. 3 and Eq. 2). For the welas 2500 sensor, $\Delta V_w$ is about $10\,\%$ of the total volume $\Delta V_{em}$ passing the OPC during the same time. For the PINE-c system equipped with a fidas-pine (fp) sensor analysing the total pump flow $F_{em} = \Delta V_{em}/\Delta t_{em}$ for particles (see Sect. 3), the INP number concentration results

from

$$n_{INP,fp} = \frac{\Delta N_{ice}}{\Delta V_{em}} = \frac{\Delta N_{ice}}{F_{em} \Delta t_{em}} \tag{6}$$

Examples from PINE-1A at higher temperatures without ice crystal formation prove that this "right edge" of the activated droplet size distribution is indeed rather sharp in typical expansion runs (Fig. A5). For data interpretation, the size threshold to distinguish between droplets and ice crystals can be set manually, however it varies with operation temperature and droplet



number concentration. Therefore, Adams et al. (in prep.) developed an automated procedure for setting this threshold. Setting this size threshold and counting all larger particles as ice crystals is a simple straightforward procedure, but neglects smaller ice particles which may also be present in the overlapping size range with the droplets and may cause an underestimation of the ice crystal number concentration. Therefore, the PINE instrument was also operated next to the AIDA cloud chamber for homogeneous droplet freezing and immersion freezing experiments in order to identify and quantify potential systematic

uncertainties and biases (see Sect. 5).

In addition to detecting the accurate number of ice crystals, the quality of ice nucleation and INP measurements also depends on measuring the precise temperature, at which the ice crystals are actually nucleated, either homogeneously or at the surface of an INP. The variability of the gas temperature in the PINE cloud chamber during 59 expansions is illustrated in Fig. 6. Here, all ice crystals detected during the 59 expansions are plotted for the relative time after start of the run in which they were

measured, and the respective gas temperature measured with three sensors located in the lower (blue), the middle (green) and the upper (red) part of the chamber (see Fig. A4). First of all, one can see that the number of ice crystals, and thereby also the number of immersion freezing INPs that caused the ice formation in these expansions, increases with decreasing temperature, which reflects the temperature dependent INP number concentration in ambient air. For individual sensors, the temperature variability from run to run is less than about $0.5\,°C$, clearly underlining that PINE is able to detect the temperature dependent

ice crystal formation from run to run at well controlled conditions. However, there is an increasing deviation of the temperature readings at the different locations in the PINE cloud chamber, with the lowest temperature measured at the bottom and the largest at the top. This inhomogeneity of the temperature across the chamber arises from the fact that there is an increasing temperature difference between the expanding gas and the almost constant wall temperature. This causes an increasing heat flux into the chamber volume and by that an increasing temperature distortion and deviation from the adiabatic temperature profile.

The hereby formed warm air tends to be collected in the top part of the chamber. The related temperature variability inside the cloud chamber impacts the temperature uncertainty for the INP and ice nucleation detection. However, it can be assumed that most of the ice crystals detected in the PINE expansion mode are formed at the lowest temperature in the bottom part of the chamber, where all the air flowing to the OPC passes through. Since ice nucleation is a strong function of temperature, it is a good first order approximation to assume the coldest temperature in the chamber to guide the ice nucleation. This assumption

will be solidified by the results of experiments presented and discussed in the following section.

An important part of PINE operations are the background runs, during which the sampled air is guided through a filter, while the operation runs are ongoing. After typically $5$ to $10$ runs, the chamber becomes particle free, as such any remaining particle counts indicate the presence of frost formation on the walls or a leak in the chamber or pipework. A typical background measurement, where almost no particles are present after 5 cycles, is presented in Fig. A3. Regular background run series are

performed with PINE at least during longer measurement phases at low temperatures. A frost-free chamber is a prerequisit for operating PINE with highest sensitivity. In case of zero background conditions, the detection limit for INP number concentrations can be calculated by dividing the minimum number of ice crystals detected in a certain volume of air. In one expansion with PINE-1A and PINE-c analysing about $0.2$ and $2$ liters of air per run, respectively, the resulting one count detection threshold is $5\,L^{-1}$ and $0.5\,L^{-1}$, respectively (see also Tab. 1). Note that the detection limit of PINE-1A is a factor of 10 lower because





only about $10\%$ of the pump flow during the expansion is analysed, whereas the OPC of PINE-c detects all ice crystals in the pump flow. If 10 consecutive runs are conducted and summed up in one hour, assuming the total run time is set to $6\,\mathrm{min}$, about 10 times more volume of air is analysed, and all ice crystals detected can be summed-up, so that the INP detection limits are reduced by a factor of 10 to $0.5\,\mathrm{L}^{-1}$ and $0.05\,\mathrm{L}^{-1}$ for PINE-1A and PINE-c, respectively, with a time resolution of one hour. When summing-up over a whole day of subsequent runs, the detection limits are further reduced to $0.02\,\mathrm{L}^{-1}$ and $0.002\,\mathrm{L}^{-1}$,

respectively.

## 5    Laboratory tests of the prototype version PINE-1A

During several test series, immersion freezing and cloud droplet freezing measurements with PINE-1A were compared to the AIDA cloud camber results. For these intercomparison studies, PINE-1A sampled aerosols directly from the AIDA chamber and was operated at similar wall temperatures as the AIDA cloud chamber. By that, the cloud expansion runs covered a similar

temperature range, and as such allowed the intercomparison of temperature-dependent INP concentrations. Figure 7 shows the results for homogeneous freezing of supercooled water droplets, which are known to start freezing in a typical AIDA cloud expansion run at about $-36\,^{\circ}\mathrm{C}$, in good agreement with other experimental results and formulations for classical nucleation theory (Benz et al., 2005). As in the experiments by Benz et al. (2005), aqueous sulphuric acid particles were first added to the AIDA chamber. Then, the aerosol particles with a diameter around $0.8\,\mu\mathrm{m}$ and a number concentration of about $200\,\mathrm{cm}^{-3}$

were sampled into the PINE-1A chamber for its homogeneous freezing experiments, followed by an AIDA cloud expansion experiment with the same aerosol. Figure 7 shows good agreement for the onset temperature of the homogeneous freezing in PINE-1A and the AIDA cloud expansion experiment. The PINE-1A data is plotted as a function of the temperature measured with the bottom temperature sensor, which always measures the lowest temperature during a run (see Fig. 6). This result underlines the assumption, that the ice formation measured with PINE is mainly controlled by the minimum temperature in the

cloud chamber.

PINE-1A was also operated next to the AIDA cloud chamber during the EXTRA18 campaign in February 2018. This campaign was mainly organized to test and calibrate the newly constructed PINE-1A in preparation to the first field campaign, which will be described in more detail in a follow-up paper. During this campaign, PINE-1A sampled aerosol particles directly from the AIDA chamber again, and measured their ice nucleation activity in the same temperature range covered by AIDA

cloud expansion runs with the same aerosols. Arizona test dust (ATD) and illite NX aerosols where used during this campaign. These aerosols are well studied for their ice nucleation activities and were also used in previous intercomparison experiments for INP instruments (DeMott et al., 2011, 2018; Hiranuma et al., 2015). We used the same aerosol sources as Steinke et al. (2015) for ATD and Hiranuma et al. (2015) for illite NX, and the methods for generating and characterizing these aerosols are described in these papers.

The supercooling or minimum temperature reached in a PINE expansion can be controlled by two parameters, the pump flow rate and the end pressure. This allows for a quick scan through a certain temperature range of ice nucleation activity. Both higher pump flow rates and lower end pressure cause a larger supercooling of the air in the cloud chamber, means a





lower minimum temperature at the end of expansion. An example is shown in Fig. 8. In this operation, PINE-1A sampled ATD aerosol directly from the AIDA chamber and measured the number fraction $f_{ice}$ of ice-active ATD particles in a series of runs starting from a temperature of about $-18\,°C$. The expansion flow rate was $5\,L\,min^{-1}$ in all runs, but the end pressure was stepwise reduced every 5 runs from about $800\,hPa$ at the beginning to about $500\,hPa$ at the end of this operation (see panel (a) of Fig. 8). This caused a stepwise decrease of the minimum gas temperature in the cloud chamber, as also shown in panel (a). The welas 2500 single particle data (Fig. 8, panel (b)) indicates an increasing amount of ice formation with decreasing minimum temperature. This stepwise increase in the number concentration of ice crystals or INPs is shown in panel (c) of Fig. 8, which depicts the time series of the ice crystal number concentrations measured at the end of each expansion.

Figure 9 depicts the ice crystal number fraction calculated with Eq. 5 devided by the aerosol number concentration for each individual run as function of gas temperature measured with the sensor in the bottom of the PINE-1A cloud chamber. The measured number concentration of ice crystals equals the number concentration of ice-active ATD particles measured in an AIDA cloud chamber experiment with the same aerosol (Fig. 9). For the PINE measurements, we estimate a temperature uncertainty of $\pm1\,°C$, mainly caused by the inhomogeneous temperature distribution in the PINE cloud chamber during the expansion run (see Fig. 6). The temperature uncertainty during AIDA cloud expansion experiments is $\pm1\,°C$. The estimated uncertainty for the ice number concentration is $\pm20\,\%$ for both PINE and AIDA, mainly due to the uncertainty in the dimension of the ODV of the welas sensor and the measured transect time of particles passing the ODV (see Eq. 1).

The same measurements as for ATD were also performed with illite NX aerosol (Figs. 10 and 11), but with both AIDA and PINE-1A starting their cloud expansions at a slightly lower temperature of about $-22\,°C$ because of the somewhat lower ice nucleation activity of illite NX compared to ATD. Within the given uncertainty ranges, the PINE-1A data is in excellent agreement with the AIDA data for both ATD (Figs. 8 and 9) and illite NX (Figs. 10 and 11). This also underlines the assumption, that the ice formation in PINE is mainly controlled by the coldest temperature in the bottom part of the chamber and that the number concentration of ice crystals, and by that the number concentration of ice-active aerosol particles in laboratory experiments and of INPs during field measurements can correctly be calculated with Eqs. 5 and 6.

## 6 Field measurements with PINE-c

We performed ground-based INP measurements with PINE-c at the SGP observatory in Oklahoma, where long-term measurements provide statistical context (www.arm.gov/capabilities/observatories/sgp). The ARM SGP field campaign, the so-called ExINP-SGP (www.arm.gov/research/campaigns/sgp2019exinp), was held from October 1 to November 14, 2019. Briefly, we have successfully operated PINE-c at the SGP site (Fig. A6) via remote control for INP concentration measurements on a continuous basis for 45 consecutive days. During the ExINP-SGP campaign, PINE-c was operated with an expansion mode time of 60 to 90 seconds, resulting in an averaged sampled gas volume of $3.7\pm0.6\,L$. This resulted in the minimum detectable INP concentration of about $0.2$ to $0.3\,L^{-1}$ for a single run of approximately 8 minutes duration. PINE-c was set to automated wall temperature control with ramping back and forth between $-5\,°C$ and $-35\,°C$ every 90 minutes, without any substantial technical issues during the whole campaign period.





Shown in Fig. 12 is the overall summary of compiled $n_{ice}(T)$ spectra measured during the ExINP-SGP campaign. Individual data points (black dots) represent 6 hours time-averaged data with a temperature interval of $1\,°C$. Here, we display the PINE-c $n_{ice}$ data for the temperature segment of $-10\,°C \geq T \geq -30\,°C$. This temperature range virtually represents the PINE-c condition, where ice nucleation through immersion freezing was warranted below local ambient dew point temperature. Any further

scientific discussions regarding PINE-c operations and observations during the ExINP-SGP campaign (e.g., deconvolution of nucleation modes, relationship between measured microphysics and local dynamics/thermodynamics, potential artefacts etc.) are beyond the scope of our current study, and will be followed up in future publications.

## 7 Summary and conclusions

We present a new instrument called PINE (Portable Ice Nucleation Experiment) for laboratory studies of ice nucleation and
field measurement of ice-nucleating particles (INPs). Inspired by the large AIDA cloud chamber (Möhler et al., 2003, 2005), the PINE instrument also uses the principle of expansion to expose aerosols from different sources to cloud-relevant conditions. By that, the sampled aerosol particles are activated to form both supercooled water droplets and ice crystals, which are detected with an optical particle counter (OPC). However, with a volume of only about $10\,L$, PINE is much smaller than the AIDA cloud chamber. The instrument is sensitive to detect ice formation and INPs in the immersion freezing, pore condensation freezing
and deposition nucleation modes in a wide temperture range from $-10\,°C$ to $-65\,°C$. Equipped with a LabVIEW control system, PINE can be operated autonomously over longer time periods and is therefore also suitable for INP monitoring at atmospheric field sites and observatories.

The operation of PINE is organized in a well defined sequence of runs. Each run is composed of three modes called "flush", "expansion", and "refill". During the flush mode, the aerosol under investigation is sampled into the pre-cooled cloud chamber.
The sampled aerosol particles are activated as supercooled cloud droplets and ice crystals during the expansion mode, depending on the pre-set wall temperature, the expansion rate and the minimum pressure reached at the end of the expansion mode. Droplets and ice crystals are detected with the OPC, and the fraction of ice-active aerosol particles or the number concentration of INPs in the sample can be calculated from the total number of ice crystals detected during the expansion mode and the volume of air that has passed the detection volume of the OPC. During the refill mode, the cloud chamber is just refilled
to the ambient pressure to immediately start the next run. In the current PINE versions, one run takes about 4 to 6 minutes, which defines the largest time resolution that can be achieved with PINE when e.g. measuring time series of atmospheric INP concentration.

Here we presented and discussed the construction and performance of both the prototype version of the new instrument, called PINE-1A, and the more advanced and commercially available version PINE-c (Bilfinger Noell GmbH). PINE-1A has a
stainless steel cloud chamber of $7\,L$ volume that is cooled with a chiller to measure immersion freezing INPs at temperatures between about $-10\,°C$ to $-40\,°C$. This instrument was tested and characterized in a series of laboratory measurements in comparison with the benchmarked AIDA chamber. PINE-1A results for both homogeneous freezing of cloud water droplets and immersion freezing of ATD and illite NX aerosols were in excellent agreement with AIDA results. The first operation





of PINE-1A during the HyICE field campaign in Hyytiälä, Finland, also demonstrated that there is only little variability of the measured droplet and ice size distribution from run to run.The INP concentration is measured with a high precision and repeatability. The temperature uncertainty is estimated to be about $\pm 1\,°C$, mainly influenced by an increasing temperature inhomogeneity during the expansion mode. The field operation also showed that the welas 2000 OPC can well distinguish between ice crystals and droplets by setting an optical size threshold, and that PINE-1A was operated over longer time periods at almost zero background conditions without any detectable frost formation on the cold cloud chamber walls. A follow-up stuy will present more results from the HyICE field activity and will discuss in more detail the performance of PINE-1A during long-term field operation.

The advanced instrument version PINE-c has a somewhat larger cloud chamber of $10\,L$ volume which is made of thin-walled aluminium and locted in an evacuated chamber for thermal insulation. The cooling system is based on a Stirling cooler and allows cooling the chamber to temperatures as low as $-60\,°C$. PINE-c was successfully operated for the first time during a field campaign conducted at the Atmospheric Radiation Measurement (ARM) Southern Great Plains (SGP) observatory in Oklahoma, USA, from October 1st to November 14th, 2019. During this field campaign, PINE was continuously operated for 45 days in a fully automated and semi-autonomous way at high time resolution of about 8 minutes with continuous wall temperature scans between $-5$ and $-35\,°C$. The overall INP concentrations ranged from about $0.2\,L^{-1}$ at $-10\,°C$ to about $200\,L^{-1}$ at $-30\,°C$. More results from this field activity will be presented and discussed in a follow-up study.

One of the unique features of PINE, in contrast to flow diffusion or mixing devices, is its operation under dry and frost-free wall conditions. Therefore, long-term continuous operation over days and weeks can be performed without the occurrence of increasing background from frost artefacts. This is achieved by drying the sampled aerosol to a frost point temperature close to the minimum wall temperature. This was proven in a series of measurements during a field campaign in Hyytiälä, Finland. The PINE-1A results form this campaign will be discussed in more detail in a follow-up publication. The sampled air needs to be humidified when its frost point temperature is clearly below the lowest chamber wall temperature. This may only be the case when sampling from extremely cold or dry environments, like polar regions or desert areas, or when sampling laboratory aerosols generated in extremely dry air. In most surface-based atmospheric sampling locations, the sample includes sufficient humidity and needs to be dried before entering the PINE chamber. Future versions of PINE may therefore also include an optional air humidification system in addition to the diffusion dryers. In addition, the newest version PINE-c is operated with a novel and liquid-free cooling system, which makes it suitable to be even operated autonomously at remote measurement sites.

Given the dearth of atmospheric INP measurements with which to challenge and inform our aerosol, cloud and climate models, an instrument, such as PINE, capable of making measurements on a routine and autonomous basis is needed. The development of PINE is timely, since INP control the radiative properties of clouds around the globe and are first order for defining cloud feedbacks (Vergara-Temprado 2018; Tan 2016). We anticipate that PINE will become a standard autonomous instrument at atmospheric observatories around the globe as well as a versatile laboratory and research tool.





## Appendix A:  Membrane diffusion dryer

The PINE instruments are equipped with a dual membrane dryer system (Fig. A1) to reduce the humidity of the aerosol sampled into the cold cloud chamber and by that to avoid frost formation on the cold cloud chamber walls. The drying efficiency of the nafion tube was measured as a function of the pressure difference $\Delta p$ between the sample flow and the counter flow and also as a function of the volumetric sample flow rate. The drying efficiency is plotted in Fig. A2 as the difference $\Delta T_d$ of the sample air dew point temperatures measured with a chilled mirror dew point sensor (MBW type 393) before and after the dryer. The measurements shown in Fig. A2 were conducted with the dew point temperature of the sample air ranging from about 6 to $7\,^{\circ}\mathrm{C}$. The drying efficiency is increasing with the pressure difference and decreasing with the sample flow rate. High drying efficiency with a drop in dew point temperature of more than $10\,^{\circ}\mathrm{C}$ is achieved when operating the dryers with a sample flow rate below 2 to $3\,\mathrm{L\,min^{-1}}$ and at the maximum pressure difference of about $800\,\mathrm{hPa}$ across the membrane.

## Appendix B:  Backgroud measurements

Operating PINE with high sensitivity for INP detection requires low or even zero background conditions. Therefore, the control system allows for regular background checks, where the instrument is set to flush mode and passing the sample flow through the bypass line with particle filter (via dashed line in Fig. 2a). A typical background run sequence (operation) from the HyICE field measurements with PINE-1A (Fig. A3) shows that the particle counts approach or drop to zero after about 4 to 5 runs. More details about background behaviour of PINE will be presented and discussed in a follow-up paper.

## Appendix C:  PINE construction and operation

Figure A4 shows the construction of the PINE-1A cloud chamber with the location of the three gas temperature sensors. For PINE measurements, a size threshold is used in order to distinguish larger ice crystals from smaller liquid water droplets in the OPC single particle data (see discussion in Sects. 3 and 4). In the absense of INPs, the droplet size distribution measured with the OPC has a sharp edge to larger particle diameters (Fig. A5), which is favorable for setting the size threshold. Figure A6 shows the first version of the PINE-c instrument in operation at the ARM SGP field campaign ExINP-SGP (www.arm.gov/research/campaigns/sgp2019exinp).

*Data availability.*  https://bwdatadiss.kit.edu/review/access/7408a0017b778cfd3131e47fc3c503f1b400f72a7a57c965979b3708dbbd93ed



*Author contributions.* O.M. wrote this manuscript; O.M. and B.J.M lead the PINE development and coordinated the laboratory and field
515 activities; M.A., L.L., F.V. conducted the laboratory experiments and field activities during the HyICE field campaign in Hyytiälä, Finland,
and analysed the respective measurements; J.N. and R.F. developed the control and data analysis software for PINE and contributed to the
analysis and interpretation of the measurements; C.B., T.P., A.H., and M.W. contributed to the engineering and construction of the PINE
instrument; H.S.K.V and N.H. conducted the measurements with PINE-c and analysed the data; all co-authors participated in the data
evaluation and interpretation and contributed in drafting this manuscript

520 *Competing interests.* The authors declare that they have no conflict of interest.

*Acknowledgements.* The development of the PINE instrument was supported by the Karlsruhe Institute of Technology (KIT) for the de-
velopment of the PINE instrument through the technology transfer project N059 PINE. We gratefully acknowledge skillful support by the
technical team at the KIT Institute of Meteorology and Climate Research (IMK-AAF), in particular by Georg Scheurig, Steffen Vogt and
525 Tomasz Chudy. We also would like to thank the organizers of and participants in the HyICE campaign from March to May 2018 in Hyytiälä,
Finland, where PINE-1A for the first time recorded field data over a longer time period of almost 2 months. We also thank the staff at the
Hyytiälä field site for their support in operatiing PINE-1A. B. J. Murray and M. Adams from the University of Leeds received financial
support through the European Research Council projects MarineIce (648661) and CountICE (862565). The material related to PINE-c is
based upon work supported by the U.S. DOE, Office of Science, Office of Biological and Environmental Research program (DESC0018979,
530 Atmospheric Processes) under Early Career Research Program Award (DE-FOA-0001761). N. Hiranuma and H. S. K. Vepuri gratefully
acknowledge the ARM-SGP technicians and administrative team as well as WTAMU Office of Information Technology for maintaining the
guest facility and supporting a remote operation of PINE-c.



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





**Table 1.** Configuration and operational parameters of PINE prototype version 1A as well as the currently available commercial version PINE-c.

|  | PINE-1A | PINE-c |
| --- | --- | --- |
| Chamber type | Stainless steel, single walled | Aluminium, thin-walled |
| Thermal insulation | 2 cm thick armaflex layer | Vacuum chamber |
| Chamber length | 75 cm | 57 cm |
| Chamber diameter | 15 cm | 18 cm |
| Chamber volume | 7 L | 10 L |
| Cooling system | Chiller Lauda (RP855) | Stirling (Thales, LPT9310) |
| Wall temperature range | 0 °C to −33 °C | 0 °C to −60 °C |
| Measurement temperature range | −10 °C to −40 °C | −10 °C to −65 °C |
| Temperature uncertainty | ±1 °C | ±1 °C |
| Wall cooling rates | $0.3\,°\text{C}\,\text{min}^{-1}$ | $0.6\,°\text{C}\,\text{min}^{-1}$ |
| Wall heating rates | $0.3\,°\text{C}\,\text{min}^{-1}$ | $0.6\,°\text{C}\,\text{min}^{-1}$ |
| Particle detector | welas 2500 | fidas-pine |
| Inlet dryer | Permapure, MD-700-24S-1 | Permapure, MD-700-24S-1 |
| Detection limit at 6 minute time resolution (single run) | $5\,\text{L}^{-1}$ | $0.5\,\text{L}^{-1}$ |
| Detection limit at 1 hour time resolution (10 runs) | $0.5\,\text{L}^{-1}$ | $0.05\,\text{L}^{-1}$ |
| Detection limit at 24 hour time resolution (240 runs) | $0.02\,\text{L}^{-1}$ | $0.002\,\text{L}^{-1}$ |





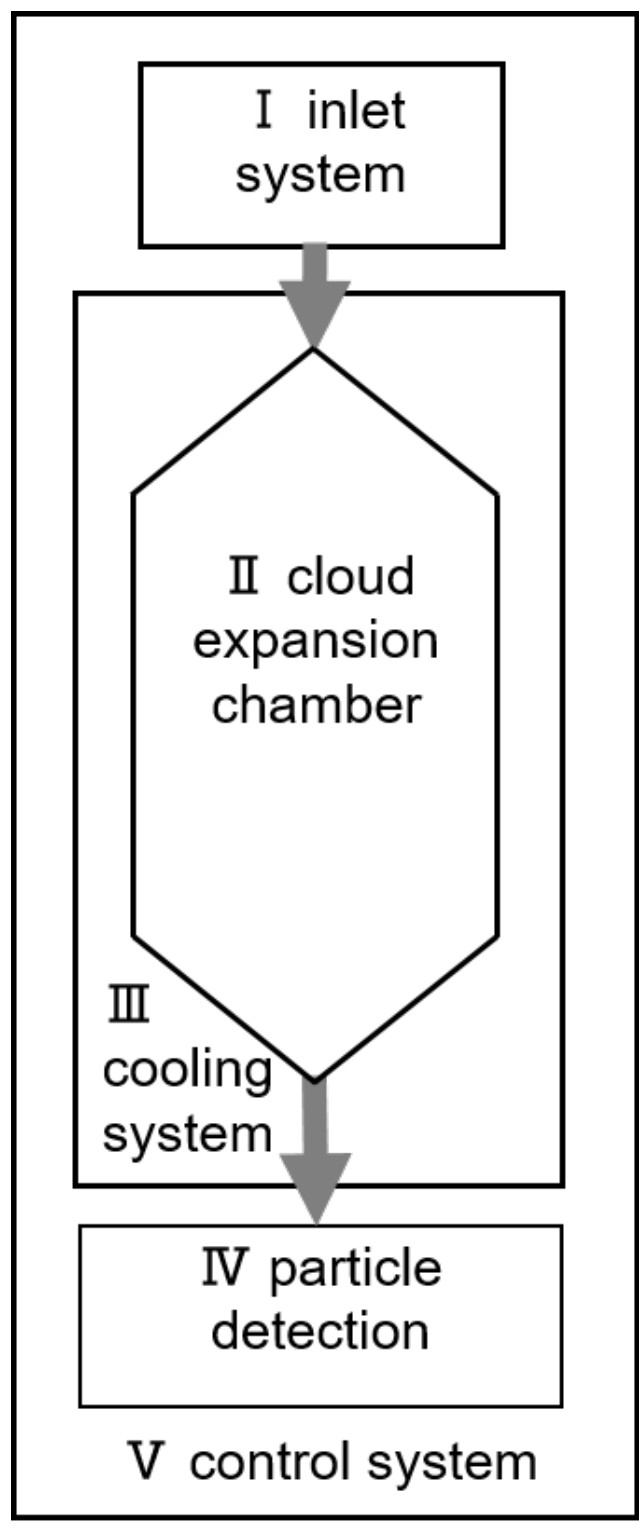

**Figure 1.** Scheme of a PINE instrument with its five basic components.





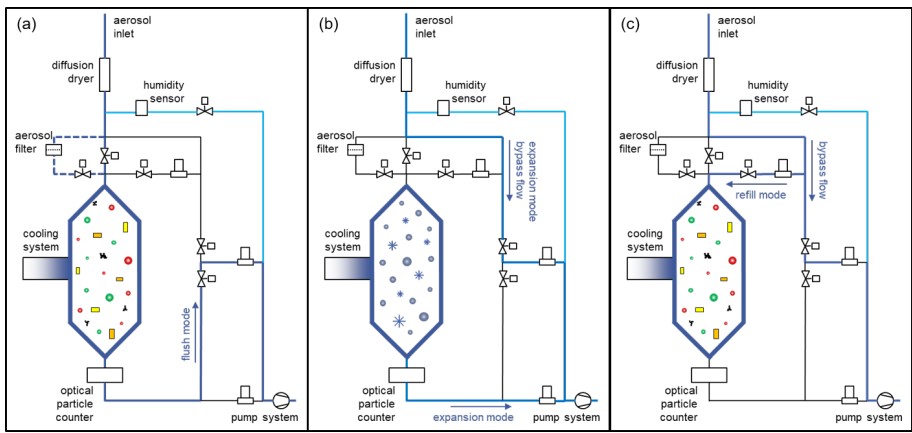

**Figure 2.** Schematic setup of the PINE-1A. The three figures show the same instrument, but in the different run modes (a) flush, (b) expansion, and (c) refill. The thick blue lines indicate which parts of the flow setup are active in the respective modes. The sampling gas flow through the humidity sensor (light blue line) is active all the time in a bypass line to the sampling pump. A background measurement can be done by passing the sample flow over an aerosol filter (dashed line, panel a). In the flush mode (a), aerosol particles are sampled (coloured various symbols), and activate into cloud droplets and ice crystals during the expansion mode (panel b, blue circles and stars, respectively). During the refill mode, aerosol particles are entering the chamber again (panel c, coloured various symbols).





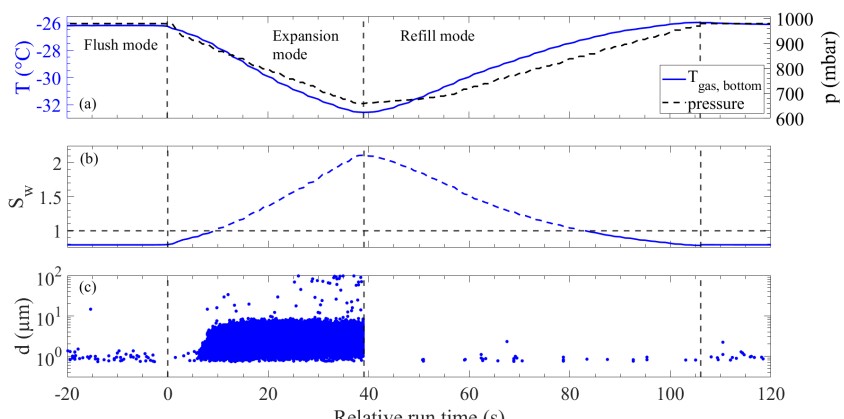

**Figure 3.** A typical run of PINE-1A showing both cloud droplet formation and ice formation during the cloud expansion mode. Upper panel: Temperature (T; blue line) and pressure (p; black line). Middle panel: Liquid water saturation ratio (Sw). Lower panel: Optical particle diameter (d) detected in the OPC. This panel shows each single particle detected by the OPC plotted as a single blue dot at the time of occurrence and with its measured optical diameter.



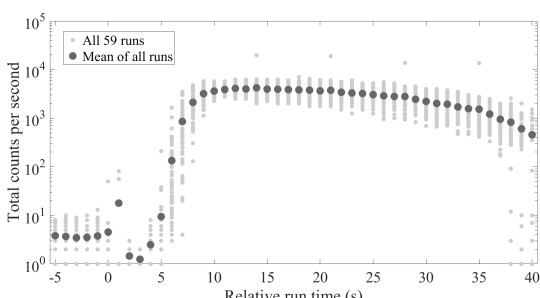

**Figure 4.** Total number counts measured with PINE-1A in 1 sec time intervals of 59 consecutive runs during the HyICE field campaign (operation 64 on 25th March 2018). The measured count rates are plotted as a function of time relative to the start of expansion. The small grey dots in this figure show the OPC count rates of individual runs, the bigger black circle the mean over all 59 runs of this operation. Before start of expansion, only larger aerosol particles are measured. The sharp increase after about 6 s of expansion is due to CCN activation of the aerosol particles in the chamber and the growth of droplets.



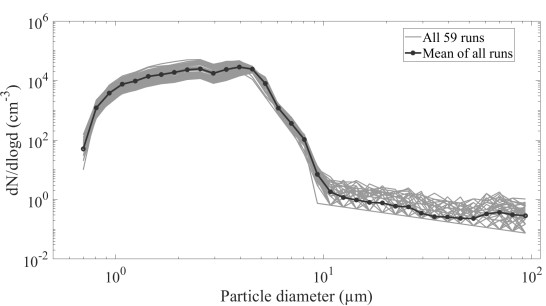

**Figure 5.** Particle size distribution for the same series of runs shown in Fig. 4.





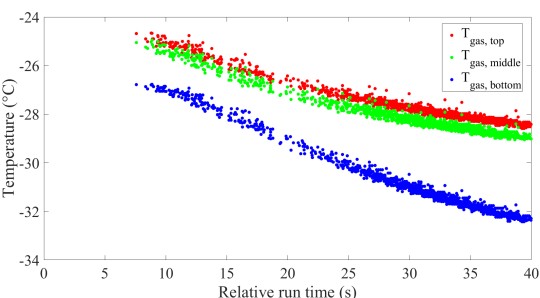

**Figure 6.** All single ice crystals measured with PINE-1A during the same operation of 59 runs shown in Figs. 4 and 5. The ice crystals are plotted for the relative time after start of the run they were measured, and the respective gas temperature measured with three sensors located in the lower (blue), the middle (green) and the upper (red) part of the chamber.



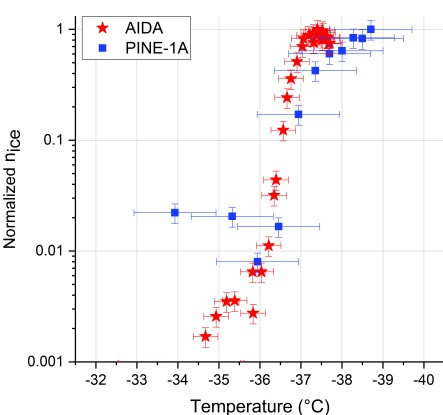

**Figure 7.** Homogeneous freezing of supercooled water droplets measured with PINE-1A and with AIDA during a PINE characterisation campaign in December 2018. For this measurement, the PINE-1A was equipped with a welas 2500 OPC and sampled sulphuric acid aerosol directly from the AIDA chamber. PINE-1A was operated at a wall temperature of about $-32.5\,°C$, the expansion run was done with a flow rate of 5 l min-1, and reached a minimum gas temperature of $-39\,°C$. The AIDA expansion was started at a temperature of about -31°C and reached a minimum temperature of about $-38\,°C$.



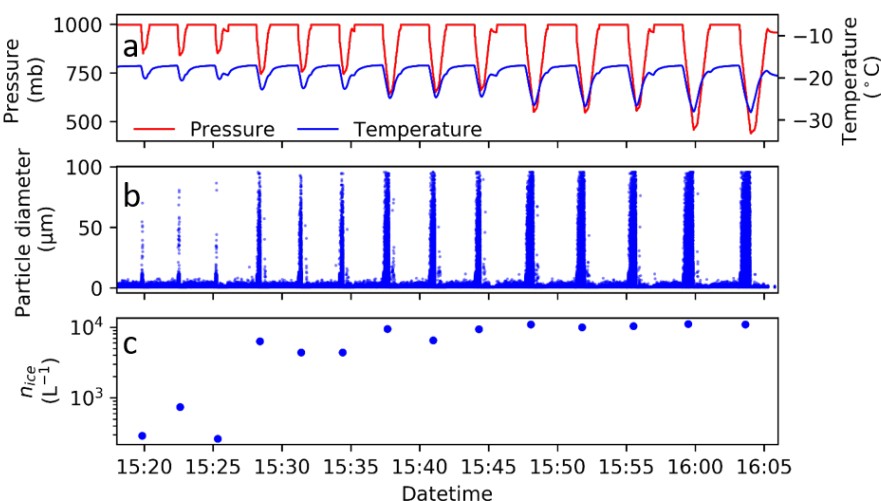

**Figure 8.** Repeated runs of PINE-1A sampling ATD aerosol from the AIDA cloud chamber during the EXTRA18 laboratory test campaign in preparation of the HyICE field campaign. The runs were started at the same temperature of about $-18\,°C$ (blue line), but the minimum expansion pressure (red line) and by that also the minimum gas temperature in the PINE cloud chamber was stepwise changed every 5th run (upper panel). Therefore, the number of ice crystals formed by immersion freezing also stepwise increased, as shown in the single particle plot from the welas 2500 OPC data (middle panel) and the ice crystal concentration measured at the end of each expansion (lower panel).

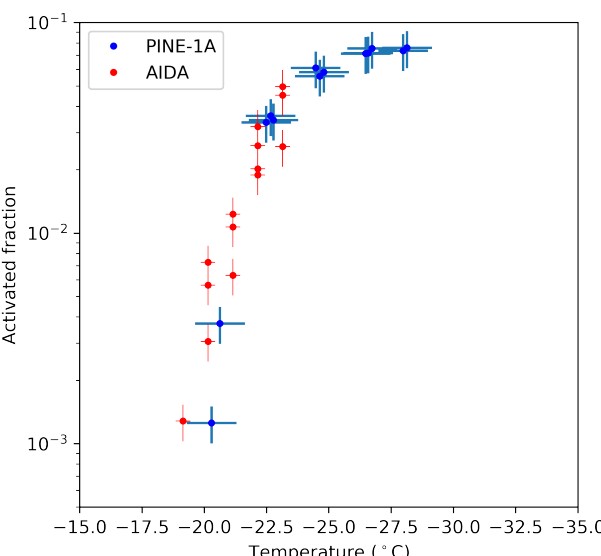

**Figure 9.** Ice-active particle fraction fice measured with PINE-1A for ATD as a function of temperature (see also Fig. 8), in comparison to fice measured in an AIDA cloud expansion experiment with the same aerosol, right after the PINE-1A runs were finished.



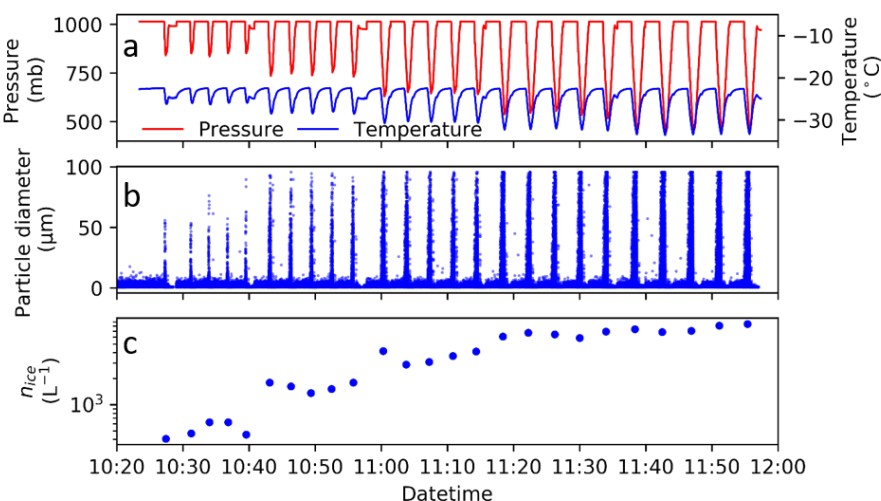

**Figure 10.** Same plot as shown in Fig. 8, but with PINE-1A sampling illite NX aerosol from the AIDA cloud chamber, and with a lower start temperature of about -22°C (see upper panel, blue line). As for ATD runs, the minimum expansion pressure (red line) and by that also the minimum gas temperature in the PINE cloud chamber was stepwise changed every 5th run (upper panel). Therefore, the number of ice crystals formed by immersion freezing also stepwise increased, as shown in the single particle plot from the welas OPC data (middle panel) and the ice crystal concentration measured at the end of each expansion (lower panel).



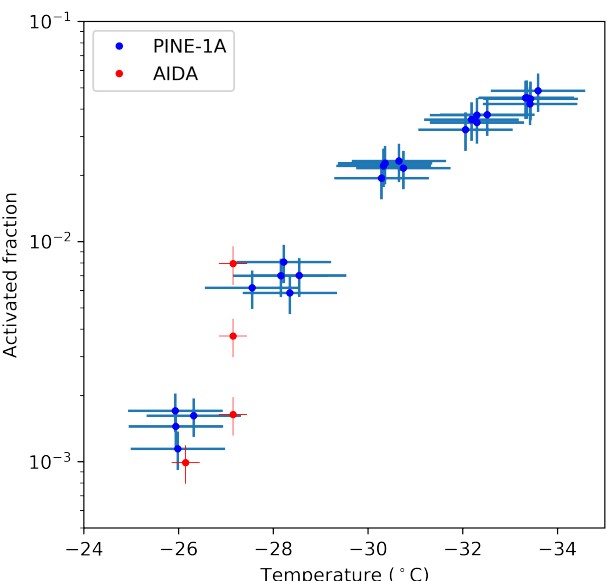

**Figure 11.** Ice-active particle fraction fice measured with PINE-1A (blue dots) for illite NX as a function of temperature (see also Fig. 10), in comparison to fice measured in an AIDA (red dots) cloud expansion experiment with the same aerosol, right after the PINE-1A runs were finished.



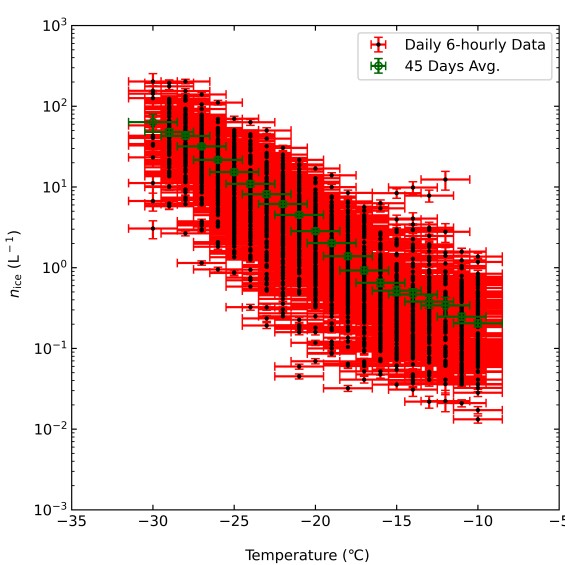

**Figure 12.** PINE-c INP concentration ($n_{ice}$) as function of the minimum gas temperature during a U.S. Department of Energy funded campaign at the ARM-SGP site in Oklahoma. PINE-c measured continuously for 45 days from October 1st to November 14th, 2019. Individual 6 hour time-averaged data and overall temperature-binned data ($\Delta T = 1°C$) are shown in black and green markers. Note the temperature uncertainty of $\pm 1.5°C$ based on the homogeneous freezing temperature calibration with ammonium sulfate aerosol particles. The $n_{ice}$ uncertainties represent relative standard errors of 6-hour averaged measurements at given temperatures.





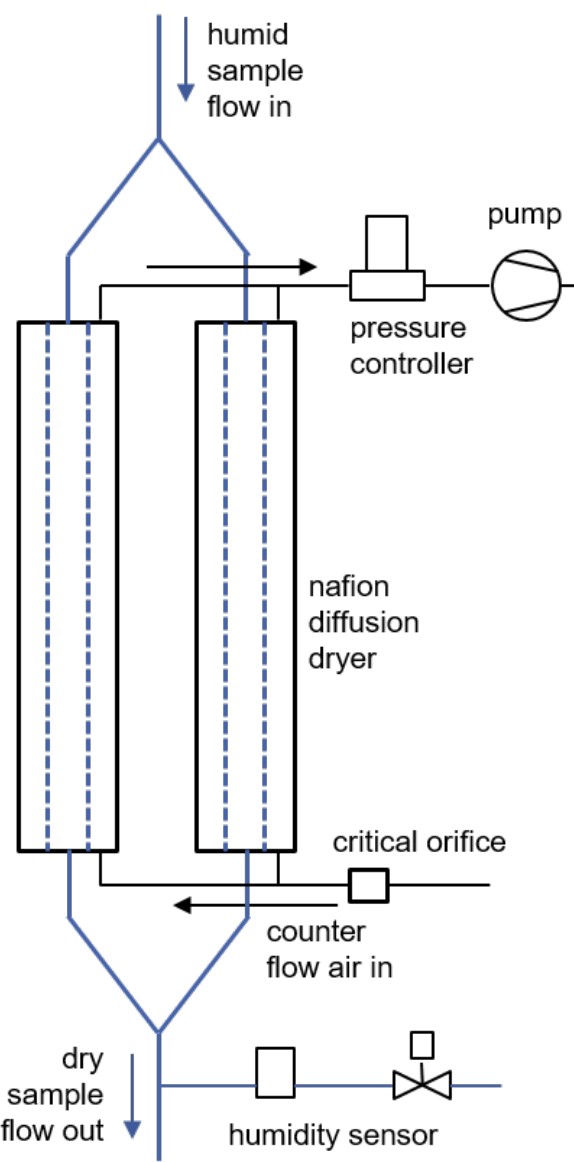

**Figure A1.** Schematic setup of the dual nafion dryer setup as part of the PINE inlet system.





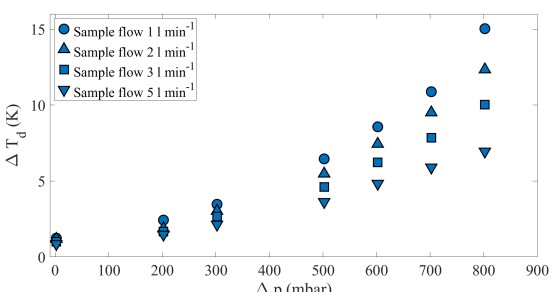

**Figure A2.** Drying efficiency of one nafion diffusion dryer, plotted as the difference $\Delta T_d$ of the dew point temperatures measured in the sample air before and after the nafion tube. The drying efficiency is increasing with the pressure difference $\Delta p$ between the sample air and the counter flow air, and decreasing with the sample flow.





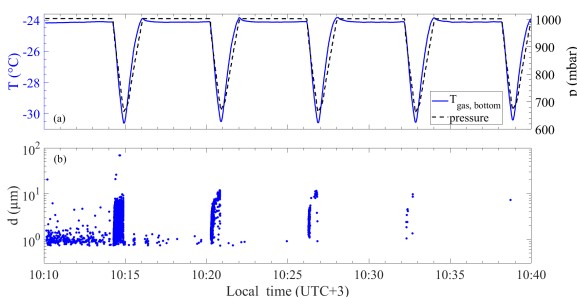

**Figure A3.** Background test run showing that after 4 consecutive expansion runs the total particle count is almost zero (only one droplet count detected in expansion no. 5).



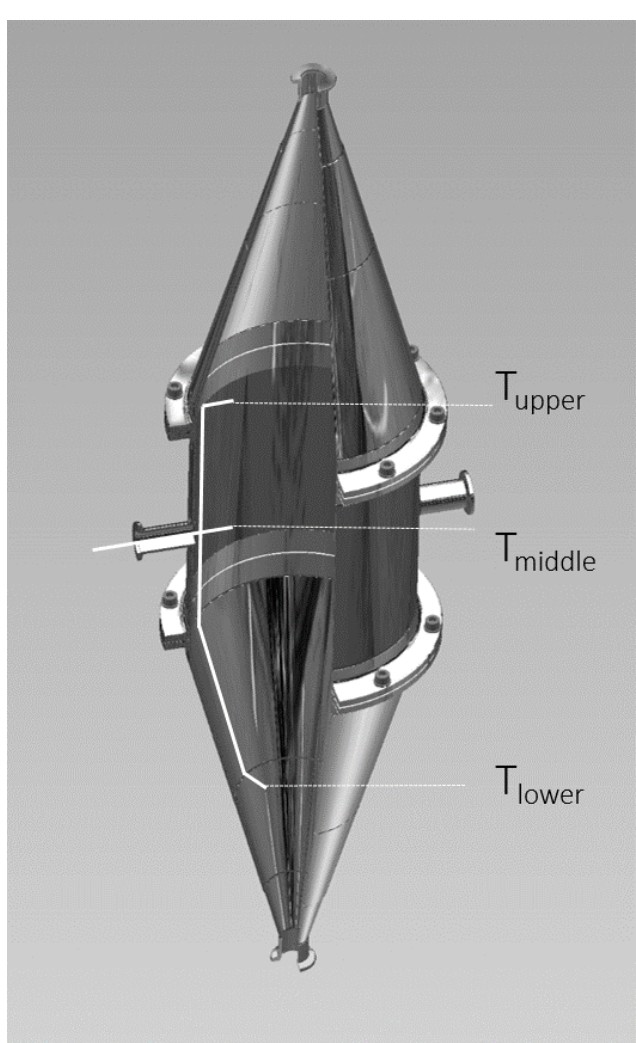

**Figure A4.** Construction of the PINE-1A stainless steel cloud chamber, without cooling and thermal insulation. The white lines indicate the location of the three thermocouples measuring the gas temperature inside the cloud chamber.





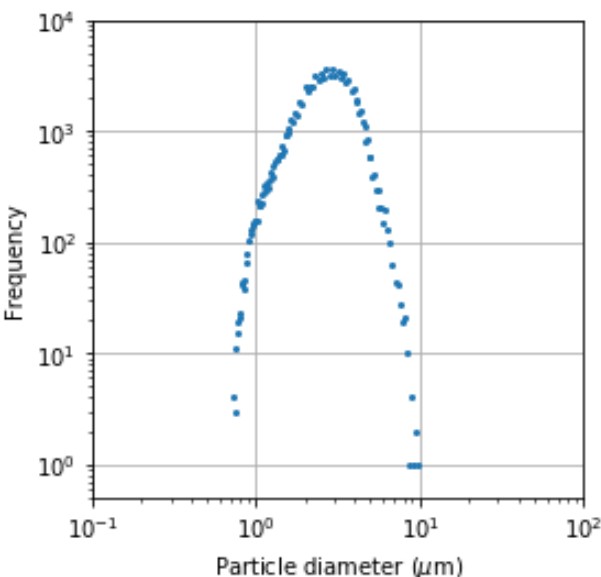

**Figure A5.** Size distribution of activated droplets measured with PINE-1A at high temperature conditions where no active INPs were present.





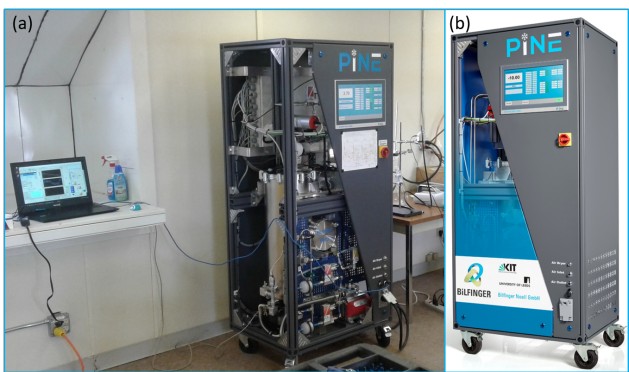

**Figure A6.** PINE-c operated at the SGP site (a) for continuous INP measurements for 45 days from October 1st to November 14th, 2019. The foto on the right side (b) shows the same instrument with all the side plates in place.