# Peer review of "The portable ice nucleation experiment PINE: a new online instrument for laboratory studies and automated long-term field observations of ice-nucleating particles"

_Atmospheric Measurement Techniques, 2020_

## Referee Comment (RC1) · Anonymous Referee #1 · 21 Sep 2020

Review of Moehler et al. "The portable ice nucleation experiment PINE: a new online instrument for laboratory studies and automated long-term field observations of ice-nucleating particles"

This manuscript details the design and performance of a new ice nucleation chamber. This instrument is based on an expansion principle, much like the AIDA chamber at KIT (location of several of the co-authors). In this regard the chamber is different than the continuous flow principle used on almost all current ice nucleation chambers. PINE therefore represents an important addition to the field.

[Figure]

The design and performance is important and the use of a long term (in this case 45 days) makes this a solid paper and very appropriate for AMT. The paper is well written and only minor revisions are needed. There are a few points I'd like to ask the authors to consider:

Starting in the Abstract but running through paper there are several unquantified terms : "...extensive.." '...good...' '...high time resolution...' These are all subjective and need to be removed.

The Introduction, although highly comprehensive, is also very long for an instrumental paper (3 pages). It seems like it could be considerably shortened.

The 'milestone' portion of the 2. Basic Principles section should be removed. It does not seem relevant to outline the timeline / dates (i.e. 20 years, first test 2016, etc.) since they don't impact the instrument performance. Please eliminate this part of the paper.

The dates of the SGP test (Oct 1 - ) is found in Section 2 and then repeated 4 times in the paper; please state once.

During HyICE, there are repeated references to CCN activation. Just as PINE is compared to AIDA, wasn't there a CCNC at HyICE? If so can the PINE droplet data be compared to those data? The topic of drop formation could be more fully developed in the paper and this would help.

Figures Please check f ice and others not in subscript ;

Figure 8 : Does "in preparation of the HyICE field campaign" have an impact on the measurement? It seems highly extraneous.

Figure 11 : Is 'aerosol, right after the PINE-1A runs were finished.' the relevant point? Is 'using the same aerosol' correct?

Figure 12 : Does not seem necessary to attribute funding to DOE here since this is

typically done in the acknowledgements. Site location seems sufficient. Inset legend – seems to mean '6 hour averaged data' (not daily)? And '45 day average'

Figure A1 : 'setup' can be removed, it is redundant after 'Schematic'

Figure A5 : Figure text appears to be of low quality and needs to be increased in resolution.

Figure A6 : 'foto' should be 'photograph'. Panel (b) appears redundant and can be removed.

---

## Referee Comment (RC2) · Anonymous Referee #2 · 23 Sep 2020

Recommendation: minor revision

A new laboratory instrument for INP measurement called "portable Ince Nucleation Experiment" (PINE) chamber is introduced in this manuscript. The design, working principles, and operational procedures of the PINE chamber are described in details, as well as preliminary results from the HyICE campaign, AIDA intercomparison and SGP-ExINP long-term measurements are provided as work cases in the paper.

The development of the PINE chamber is a great contribution to the INP research

field in specific and the Atmospheric science in general. It also provides long-term monitoring capability to operation-oriented organizations. The topic fits AMT scope perfectly. The paper is well organized and written. After the authors address my minor points, it should be in good shape for publication on AMT.

Minor comments:

The font on many figures is too small.

Line 175: remove "and"

Line 265: What is the aerosol size range for these concentrations?

Lines 271 to 275: Is the assumption of ice saturated condition at the beginning of expansion reasonable? The response of the OPC does agree with this assumption. But is it universally valid?

Line 287: "larger than the dense"

Figure 6: Can a turbulence be introduced to the chamber to mix the air so that the temperature is more uniform across the chamber?

Line 446: replace "largest" with "highest".

Line 468: Based on Fig. 12, the minimum INP concentration is about 0.02 L-1, not 0.2 L-1.

There are multiple places stating that details on HyICE results and SGP-ExINP results will be discussed in details in future papers. Can reduce the redundancy.

---

## Referee Comment (RC3) · Paul DeMott (Referee) · 29 Sep 2020

**General Comments** As a scientist in this research area who both continues strong collaborations with some of the group represented on this paper and has promoted use of continuous flow diffusion chambers (CFDCs) for ice nucleation measurements over many years, I figure it is important to self-identify in this review. In this manuscript, the PINE instruments are introduced, appearing to represent a great new tool for the community, and with promise for meeting certain INP monitoring and experimental study needs. The new results for unhindered immersion freezing are very encourag-

ing, admirable in being achieved over a relatively short period of development. From the standpoint of a publication documenting a new method, there were a few things missing for me as a very interested reader. Hence, I list a number of specific comments/questions below, driven by my desire to understand the instrument clearly. In short summary, 1) there was not a full description of principles and device(s) in order to understand what challenges may be met in applying the method for the range of ice nucleation studies inferred to be possible (deposition and immersion freezing to $-60°$C (i.e., only immersion freezing is thus far discussed to the point of homogeneous freezing conditions); 2) uncertainties were given relatively limited discussion (especially at the limit of detection); 3) there was no discussion of consistency of results with physical expectations that might be revealed from, for example, microphysical modeling considerations; 4) relatedly there seemed more cursory consideration given to defining the relevant temperature associated with a measurement (I did follow the arguments, although the confirmation was mostly by comparing to AIDA), the role of growth time and sedimentation if any, clear separation of water and ice given that the latter occurs usually a few orders of magnitude lower than the water drop concentrations; and 5) finally, the introduction of field data and field instrument was rather abbreviated considering the nature/nuances of that application and considerations that will impact operation across the stated T and supersaturation range of the device in the presence of varying atmospheric conditions and full aerosol distributions. The field data only serve the purpose of demonstrating a range of data collected during automated operation for a period, as there is no other discussion of the data provided. I expect that some of the lack of clarity that I sensed will be resolved in review here. I understand, of course, that full information on any new device is often revealed over some time, often in a number of separate publications. This is clearly underway as indicated by a paper in preparation (and other intercomparison studies I am aware of), but it suggests then that some of the statements herein may require a few caveats because supporting data are not fully shown. Hence, I might even suggest consideration of a title change to include something like "An introduction..." or "A first evaluation..." or "First description and

results from. . ." or something to that effect. That is not an adamant request, simply a suggestion. The paper is otherwise well-written and an anticipated and welcome addition to the literature.

**Specific Comments**

Introduction:

1) The introduction was comprehensive, perhaps overly so for a paper describing a new instrument. It was long, and not so much related to the development itself other than attempting to meet motivations.

2) Lines 87-89: It seems clear that both low and high time resolution are desirable things for different scenarios. High time resolution is arguably not useful if one is attempting to document the most special INPs, the ones that even the PINE instrument may struggle to measure in all but INP-rich environments such as shown later in this paper. I see higher time resolution measurements as highly useful, but not sufficient, unless their resolution can match higher volume collections. Some of the studies referenced to preface this statement were made with instruments capable of even higher time resolution than the PINE, but the issue I am speaking of is resolving low INP concentrations in some environments. Those other methods have been developed for automation as well, a point I will raise next.

3) Lines 97-100: I would say to be fair that these statements need modification or qualification for other studies in the recent literature. I think that the continuous flow chamber developments reflected in Bi et al. (2019) and Brunner et al. (2020) meet the criteria of operating more than "periodically" and of saving "intensive man-power and time for operation or offline analysis." Such developments are advancing rapidly across the discipline. Those instruments also appear to be capable of higher time resolution than is demonstrated for the PINE instrument in this paper. It is also implied here that time resolution of minutes will somehow solve the INP size and chemistry resolution issue, although due to statistics (sample volume and particle numbers) it

is hard to imagine this as yet being achievable for single INPs except in high loading situations. Rather, this would occur by correlation to independent high resolution composition measurements for all aerosols, which sometimes does not work for comparing directly to specialized INPs. Hence, I see high resolution INP capabilities as one piece needed in the course of a full development.

4) Line 103: The stated temperature range is what the instrument is designed for, but no exploration of capabilities to make useful measurements to as low as $-60°$C are given in this manuscript. It appears as a potential capability, only in that the temperature can be achieved in PINE-c. One can imagine that challenges in operating and interpreting data to that lower limit could be significant (e.g., control on final T and RH of expansion, low water vapor pressure and slow ice crystal growth rates), and not simply depend on the capacity of the cooling system (line 121). I suggest to stick to what is demonstrated in this paper, as far as confirmed operational capabilities and to clearly identify capabilities that remain to be defined.

Basic principles and milestones of the PINE development:

1) Line 121 repeats the assertion that likely requires "potential" as a caveat. No low temperature data are shown excepting the homogeneous freezing onset for grown droplets.

PINE instrument setup:

1) Line 169 or thereafter: Have particle losses been characterized through the nafion dryer system? For that matter, I realize in reviewing these comments that particle transfer versus size into the PINE systems has not been discussed.

2) Lines 172-173: Perhaps this is irrelevant since an aircraft system is not yet described, but I wondered about the use of the nafion system on aircraft where the pressure drop will be limited at higher altitudes. Will the system work over the needed ranges in this scenario?

[Figure]

3) Lines 177-180: I am curious about the later tests shown for background, simply because I did not understand the implications of no background particles found after five runs. Why does it take five runs to decrease, and does it mean that any background is then absent from thence forward in time? Have you explored this systematically, and/or after hours of operation? My personal understanding from an overlapping study in time with the one at the SGP site, is that the dewpoint was $-10°$C in that case, and that background counts at some level were always detected, if minimal. Hence, the basic question is if it is understood what dewpoint is sufficient for frost-free operation at any given T?

4) Lines 183-184: It was not clear to me what actually constitutes the cooling system? Is it a plenum around the chamber and this is fed by the large chiller reservoir?

5) Line 191: I expected that the minimum air temperature achieved would be colder than the minimum cooling temperature? Why are they the same?

6) Line 195: Can you explain the Stirling cooler method of cooling the wall of PINE-c for those of us unfamiliar with the exact cooling mechanism? E.g., fluid versus expansion cooling or whatever it is. The details on cooling systems in general does not match the later attention to detail of the OPCs.

7) Line 198-200: Again, the cooling is understood, but the utility for performing low temperature ice nucleation experiments, especially where this will presumably involve more special control over the expansion conditions to meet some final peak relative humidity, is not yet discussed or demonstrated herein.

8) Line 222: Does this more limited volume used to define ODV explain the higher value of lowest detection limit concentration listed in Table? Perhaps worth noting here, since It only comes up again at the end of section 4.

PINE operating principle:

1) Line 250: To this point, the definition of ice crystals versus drops has not been made.

Perhaps add a short note about this, "...as discussed later in this section"? Otherwise, this raised a number of questions immediately.

2) Lines 269-270: Figure 3 is an important figure, and it raises a number of questions that were mostly answered in time over this section. However, I will list a number of them here. Immediately I wondered why the lowest temperature measured was used. As an aside, this point (lowest T used) should also be stated in the figure caption, for clarity. What differences are seen in these temperatures, and what uncertainty does this create? Are concentrations referenced to the entire integrated time interval and volume of expansion (and will this be the case also for the PINE-c), and do they represent the lowest temperature achieved (e.g., there is a $4°$C cooling shown in the figure over the time of the expansion)? Hence, is it one measurement or many, and how are the sub-intervals defined? A range of apparent ice crystal sizes are shown in Fig. 3, up to 100 microns. Are these ice sizes consistent with expectations of grown sizes for the conditions and growth times? The PINE chamber is quite small compared to the AIDA chamber where volumetric concentrations are assessed in situ. Is there sedimentation that could impact inferred concentrations and their reference temperature for the smaller geometry of the PINE? Have any such calculations been made at this time, or are they planned?

3) Lines 273-275: Regarding the starting vapor saturation ratio for expansion, you assumed this or you set that partial pressure based on a room temperature RH measurement? Why would it be ice saturated if there is no ice on the walls? Or is it close enough as determined on some other basis? This would seem important for future use toward other measurements than immersion freezing.

4) Line 288: Here an important distinction may arise, but perhaps the authors can correct any misconception I have. While described as purely immersion freezing, the temperature is already cold at the point of expansion, and so does the measurement also not integrate some proportion of INPs from any/all INP mechanisms, other than contact freezing, that ensue as the air rapidly cools and ultimately exits the chamber

through the OPC? That is, somewhat similar to CFDCs when they are operated for bringing air to a final RH that is well above water saturation?

5) Line 317-318: Concerning addressing the size threshold setting for ice crystals, I struggled a bit to reconcile Figures A5 and 5. In A5, the scale is frequency, and it spans about three orders of magnitude out to 10 microns. Is there an issue in the fact that if cloud droplet concentrations range up to 1000 per cubic centimeter, and activated INP concentrations could range down to 1 per liter, then assessment of cloud drop frequency would have to be made over a greatly extended period of time to capture the tail of the distribution? Or is it simply the case that repeated experiments like the one in Fig. A5 never indicated a drop even in the size range greater than 10 microns? It might help to add the time and/or volume of assessment represented in Fig. A5. Clearly, Fig. 5 shows particle numbers appearing in these larger size ranges at 4-5 orders of magnitude below cloud droplet concentrations at least. This is an issue that perhaps deserved more attention in the paper, but if I understand, sensitivities of the ice cut size threshold will be more extensively covered in Adams et al. It would be good to add a reference for that paper, if it is now in submission.

6) Lines 348-349 and lines 359-360: Note that the first statement repeats from earlier in the manuscript. One example is provided in Fig. A3. Perhaps repeating myself also, is this the very start of operations, or a period during the midst of operations? Why does it take 5 cycles at all, and does the background then stay that low in all cases? What does this depend on? The question arises again in the later sentence where long time operating detection limits are listed. Do not these very low detection limits listed for long operation imply the need for validating backgrounds being below such levels over such long times?

Laboratory tests of the prototype version PINE-1A:

1) Lines 374-375: In Fig. 6, there looks to be up to 1C temperature uncertainty in defining the lowest temperature attributed to ice nucleation. Since T is not spatially

uniform in the chamber, do you anticipate a bias in sampling only part of the flow as in PINE-1A versus all of the flow in PINE-c? Also, Figure 6 and its caption could use a little attention to description. At present the data are described as "all single ice crystals measured." Should it say something like "Data points indicate all single ice crystal event temperatures..."?

2) Line 391: Just a note that there seems an inconsistency between the statement of a minimum pressure reduction every 5th cycle versus what is shown in Fig. 8 (and stated in that caption). It looks like 4 cycles. It is 5 cycles in Fig. 10.

3) Lines 401 to end of section: The basic agreement shown between AIDA and PINE in Figures 7, 9 and 11 (over a more limited range) is excellent. I again wonder here about the percentage uncertainties being constant over the entire dynamic range of ice concentrations. For example, at the LOD, the true uncertainties must be larger, no? That statistical uncertainty does not appear to be captured in defining uncertainties based on the OPC ODV alone. I guess I expected based on statistical count considerations that the uncertainties should be larger for lower INP concentrations. Additionally, given that ice concentrations are integrated over the range of temperatures present throughout the volume, and if some of the crystals grow in that time to 50-100 microns (would be good to state the typical mode size), does sedimentation assuredly not impact/skew the results attributed to one temperature? There could be differences as to how this is measured temporally in situ in AIDA versus drawing the entire tank flow from the PINE, and there is some room for not discerning that in the comparisons shown. Nevertheless, a minor point overall.

Field measurements with PINE-c:

1) Lines 416-417: With an expansion mode time of 60-90s, a question arises as to the applicability of the discussion of temperature attribution and method for calculating INP concentrations with the PINE-c versus PINE-1A. Were they exactly the same (lowest T used, etc) for these presented analyses?

2) Fig. 12: This is a nice compilation of results, if leaving a lot of room for discussion of their meaning still (i.e., variability of 2 orders of magnitude temporally at any T,) and raising all of the questions listed in the last sentence of this section. It is a minor concern for showing them in this manner, simply as a demonstration that the data were collected more or less autonomously over this period (maintenance or other attention needed were not discussed). Let me ask one thing though. The flattening of the INP concentrations toward the higher temperature limit of detection is interesting, but raises a question regarding the confidence in these results. The uncertainties are based on relative standard errors. The percentage errors are quite small and I wonder how these can be the same at the LOD as they are at any other conditions. This is the same question raised for PINE-1A.

3) Lines 423-424: I am not sure what is meant by "warranted" here. Possible? Also, can the point regarding the dewpoint temperature be clarified? Dewpoint is not controlled somehow? It would be much higher in summer and much lower in winter. How might this affect the operational range, background etc, or does this remain to be investigated?

4) Lines 425-427: What exactly is meant by deconvolution of nucleation modes? Meaning different operation of the PINE than discussed in this paper, which is immersion freezing? Or meaning resolving what I mentioned earlier in this review, the temporal evaluation of data during single expansions? This is a point that should be clarified, as it is important to state which potential aspects of PINE measurement capabilities are demonstrated in this first publication and which remain.

**Other editorial comments:**

Line 96: typo – based

Line 266: Suggest "one of" after "An example of…"

Line 282: "so" not needed before "calculated"

Line 287: Suggest "than" for "as"

Figure 3 caption: Suggest to add "Calculated" at start of sentence starting "Liquid water…"

**References:**

Bi, K., G. R. McMeeking, D. Ding, E. J. T. Levin, P. J. DeMott, D. Zhao, F. Wang, Q. Liu, P. Tian, X. Ma, Y. Chen, M. Huang, H. Zhang, T. Gordon, and P. Chen, 2019: Measurements of ice nucleating particles in Beijing, China. Journal of Geophysical Research: Atmospheres, 124, 8065–8075. https:// doi.org/10.1029/2019JD030609

Brunner, C. and Kanji, Z. A.: Continuous online-monitoring of Ice Nucleating Particles: development of the automated Horizontal Ice Nucleation Chamber (HINC-Auto), Atmos. Meas. Tech. Discuss., https://doi.org/10.5194/amt-2020-306, in review, 2020.

---

## Referee Comment (RC4) · Anonymous Referee #4 · 7 Oct 2020

The knowledge of ice nucleating particles and their impacts on clouds was restricted by the development of measurement techniques and instruments. This manuscript presents a new instrument based on expansion chamber for both laboratory studies and field observations to measure ice nucleating particles. The authors successfully demonstrate the applicability of their new instrument to be compared with AIDA and deployed in a field campaign. Different from the commonly used Continuous Flow Diffusion Chamber (CFDC), PINE is truly the first commercial instrument capable of automated long-term continuous observation, and its development provides an excellent

complement to enrich measurement methods. This research aligns well with the scope of AMT. The manuscript is well written and easy to follow, thus should be acceptable for publication after considering following minor comments:

P2 Line28: The abbreviation "(INPs)" should not be linked directly after "...atmospheric aerosol particles". Moreover, two sets of parentheses are used instead of one. For example, "...atmospheric aerosol particles (INPs) (Vali et al., 2015)".

P2-3: The need for promoting INP monitoring was put forward until P3 Line 63, and the new methods and instruments for INP measurements were discussed from P3 Line 90. Please consider adding more descriptions and comparisons of existing instruments, especially the most commonly used CFDC, and simplifying the context on INP.

P9 Line275: A parenthesis is missed.

P12-14: PINE-c performed the field measurements to demonstrate its capability, however, the comparisons with AIDA and performance tests were conducted by the prototype version PINE-1A. PINE-c is a further developed version with major upgrades in chamber type, cooling system, controlled temperature range, particle detector, and so on. So direct characterizations and tests of the performance of PINE-c would be helpful.

Figure 7: Please notice the superscript of the unit, "...5 l min-1".

[Figure]

---

## Author Comment (AC1) · 10 Dec 2020

**Author comment in response to the comments provided by Referee # 1**

We thank referee # 1 for her/his effort in reading and commenting our manuscript. In the following, we report the referee's comments (in italics), give point-by-point answers, and suggest manuscript revisions based on the referee's comments and our answers. Respective reference will be given to the line numbers of manuscript version 1.

**Referee comment:** This manuscript details the design and performance of a new ice

nucleation chamber. This instrument is based on an expansion principle, much like the AIDA chamber at KIT (location of several of the co-authors). In this regard the chamber is different than the continuous flow principle used on almost all current ice nucleation chambers. PINE therefore represents an important addition to the field. The design and performance is important and the use of a long term (in this case 45 days) makes this a solid paper and very appropriate for AMT. The paper is well written and only minor revisions are needed. There are a few points I'd like to ask the authors to consider: Starting in the Abstract but running through paper there are several unquantified terms : "...extensive ...", "...good ...", "...high time resolution ...". These are all subjective and need to be removed.

**Answer:** We scanned the manuscript for such unquantified terms and removed most of them (see lines 9, 15, 61, 297, 493) or replaced them with more quantitative statements (see lines 11, 371).

**Referee comment:** The Introduction, although highly comprehensive, is also very long for an instrumental paper (3 pages). It seems like it could be considerably shortened.

**Answer:** The third referee (Paul DeMott) also mentioned the introduction to be overly comprehensive. We agree and suggest removing or shortening the following parts:

Remove section about cirrus clouds (lines 37 to 43) which are not further subject of this paper. We had included this short paragraph in the first manuscript version because PINE is also capable of measuring INPs in the cirrus cloud temperature regime. This is subject of ongoing activities and the further development of the PINE instrument.

Reformulate lines 56 to 61: "Existing parameterizations are applied in models to calculate and predict primary ice formation in clouds, however, the atmospheric INP data that we can compare with global fields of model predicted INP concentrations are limited in spatial, ...

Remove lines 69 to 76 ("While high temperature INPs ... may dominate high temperature INPs in many regions (. . . )."

Modify line 77 to "Most of previous INP measurements were only sensitive to immersion freezing . . . ".

Modify lines 103/104 to ". . . in a wide temperature range. In this paper, PINE's ability is demonstrated to measure in the mixed-phase cloud temperature regime from $-10\,°C$ to $-40\,°C$. PINE is also able to measure ice nucleation at cirrus cloud temperatures to about $-65\,°C$, which is the topic of ongoing studies."

**Referee comment:** The 'milestone' portion of the 2. Basic Principles section should be removed. It does not seem relevant to outline the timeline / dates (i.e. 20 years, first test 2016, etc.) since they don't impact the instrument performance. Please eliminate this part of the paper.

**Answer:** We believe that the experience in operating the AIDA cloud expansion chamber was indeed an important contribution to the PINE development. Referring to AIDA operation as a well-known and well-cited cloud simulation chamber also introduces the reader to the basic operating principles of PINE. Moreover, PINE was developed during relative short time, as e.g. compared to the development of continuous flow diffusion chambers. Thus, we believe that it is interesting to the reader to be informed about the development steps. Therefore, we like to keep this part, but have made the following modifications:

Change first sentence of section 2 (lines 111/112) to "The idea for PINE resulted from the experience in operating the AIDA facility for cloud experiments at simulated conditions of up-drafting atmospheric air parcels."

Modify the last two sentences of first paragraph of this section (lines 122 to 124) to: "Large aerosol particles, droplets and ice crystals are measured and counted with an optical particle counter (OPC). Placing the OPC in the vertically oriented pump tube below the cloud chamber was one of the critical development ideas for PINE (see

patent applications DE 10 2018 210 643 A1 and US2020/0003671 A1). PINE can be operated both for ice nucleation research in the laboratory, and for INP measurements in field campaigns or long term monitoring activities.

Remove the sentence "This setup was operated in a cold room . . . attached to the pump line (see patent applications . . . )" (lines 127 to 130).

**Referee comment:** The dates of the SGP test (Oct 1 - ) is found in Section 2 and then repeated 4 times in the paper; please state once.

**Answer:** Done as suggested (see lines 414, 466).

**Referee comment:** During HyICE, there are repeated references to CCN activation. Just as PINE is compared to AIDA, wasn't there a CCNC at HyICE? If so can the PINE droplet data be compared to those data? The topic of drop formation could be more fully developed in the paper and this would help.

**Answer:** We were using the term CCN activation because aerosol particles sampled into the PINE chamber first act as cloud condensation nuclei to form supercooled droplets which then eventually freeze when they include an ice-active aerosol particle (INP) at the given temperature. In other words, the same particle acts as CCN and INP. However, CCN activation happens in a fast cloud expansion process without independent control of the relative humidity or supersaturation. Therefore, an expansion cloud chamber like PINE or AIDA cannot control or quantify the CCN process as a function of relative humidity, just is then filled with the droplets resulting from the CCN activation and diffusional droplet growth processes, but without quantifying the parameters controlling those processes. The size distribution of the resulting droplet cloud can then well be measured with the OPC and compared for subsequent runs, as demonstrated by first field measurements with the PINE-1A prototype instrument during the HyICE field campaign. To clarify this, we suggest the following changes to the manuscript line 296: "This means that PINE is able to reproduce the formation of the supercooled droplet cloud in repeated runs at constant sampling and operation conditions, . . ."

**Referee comment:** Figures Please check f ice and others not in subscript ;

**Answer:** checked and corrected

**Referee comment:** Figure 8 : Does "in preparation of the HyICE field campaign" have an impact on the measurement? It seems highly extraneous.

**Answer:** Yes, agreed and removed.

**Referee comment:** Figure 11 : Is 'aerosol, right after the PINE-1A runs were finished.' the relevant point? Is 'using the same aerosol' correct?

**Answer:** Yes, correct, both chambers are using the same aerosol. Therefore, no modification needed here.

**Referee comment:** Figure 12 : Does not seem necessary to attribute funding to DOE here since this is typically done in the acknowledgements. Site location seems sufficient. Inset legend seems to mean '6 hour averaged data' (not daily)? And '45 day average'

**Answer:** Reference to DOE funding will be removed. The figure indeed shows temperature binned data for both 6 hour averaged data and average over all 45 days. We will change the legend and rephrase the figure caption as follows: "Temperature-binned concentrations data ($\Delta T = 1\,^{\circ}$C) is shown for 6 hour time averaged data (black markers) and 45 days averaged data (green markers)."

For consistency with the updated Figure 12, we also suggest modifying lines 15/16 of the abstract as follows: "...with continuous temperature scans for INP measurements between $-10\,^{\circ}$C and $-30\,^{\circ}$C ."

**Referee comment:** Figure A1 : 'setup' can be removed, it is redundant after 'Schematic'

**Answer:** done

[Figure]

**Referee comment:** Figure A5 : Figure text appears to be of low quality and needs to be increased in resolution.

**Answer:** Figure with higher quality will be included in the revised version of the manuscript.

**Referee comment:** Figure A6 : 'foto' should be 'photograph'. Panel (b) appears redundant and can be removed.

**Answer:** "foto" changed to "photograph". Because the left shows a photograph of PINE-c at the SGP field site, and the right the 3-D construction of the commercial PINE version as a new instrument, we would like to keep the figure as is, but suggest to change the caption as follows: "Photograph of PINE-c (a) located at the ARM-SGP site in Oklahoma for 45 days of continuous INP measurements from October 1st to November 14th, 2019. Part (b) on the right shows a composite photograph of the same instrument before delivery."

---

## Author Comment (AC2) · 10 Dec 2020

**Author comment in response to the comments provided by Referee #2**

We thank referee #2 for her/his effort in reading and commenting our manuscript. In the following, we repeat the referee's comments (italics), give point-by-point answers, and suggest manuscript revisions based on the referee's comments and our answers. Respective reference will be given to the line numbers of manuscript version 1.

**Referee comment:** *A new laboratory instrument for INP measurement called "portable*

[Figure]

*Ince Nucleation Experiment" (PINE) chamber is introduced in this manuscript. The design, working principles, and operational procedures of the PINE chamber are described in details, as well as preliminary results from the HyICE campaign, AIDA intercomparison and SGP-ExINP long-term measurements are provided as work cases in the paper. The development of the PINE chamber is a great contribution to the INP research field in specific and the Atmospheric science in general. It also provides long-term monitoring capability to operation-oriented organizations. The topic fits AMT scope perfectly. The paper is well organized and written. After the authors address my minor points, it should be in good shape for publication on AMT.*

*Minor comments: The font on many figures is too small.*

**Answer:** This was also mentioned by Referee #1. We will re-plot the figures with larger fonts.

**Referee comment:** *Line 175: remove "and"*

**Answer:** done

**Referee comment:** *Line 265: What is the aerosol size range for these concentrations?*

**Answer:** The majority of particles was smaller than $500\,\mathrm{nm}$ in diameter. Only a minor fraction ($< 0.1\,\mathrm{cm}^{-3}$) had diameters between $0.5\,\mathrm{\mu m}$ and $5\,\mathrm{\mu m}$. We will add the following to line 264: "..., with the majority of particles smaller than $0.5\,\mathrm{\mu m}$ in diameter, ..."

**Referee comment:** *Lines 271 to 275: Is the assumption of ice saturated condition at the beginning of expansion reasonable? The response of the OPC does agree with this assumption. But is it universally valid?*

**Answer:** Yes, we think this assumption is reasonable because the frost point temperature of the air sampled into the chamber was higher than the average wall temperature. The excess water vapor quickly condenses to the cold chamber walls, so that ice saturated conditions of the air inside the cloud chamber are reached. We re-phrase line 273 for including this information.

**Referee comment:** *Line 287: "larger than the dense"*

**Answer:** changed

**Referee comment:** *Figure 6: Can a turbulence be introduced to the chamber to mix the air so that the temperature is more uniform across the chamber?*

**Answer:** This is a good idea and suggestion we have already discussed among the PINE team members. Of course, such fan driven mixing is an important part for operating the large AIDA cloud chamber at homogeneous gas temperature conditions. Until now, we did neither test this option for the small 10 L PINE chamber nor did we discuss the technical solution. But it is an option for further developing and improving the PINE chamber.

**Referee comment:** *Line 446: replace "largest" with "highest".*

**Answer:** done

**Referee comment:** *Line 468: Based on Fig. 12, the minimum INP concentration is about 0.02 L-1, not 0.2 L-1.*

Answer: The referee is right, it will be corrected. Thanks for noting this.

**Referee comment:** *There are multiple places stating that details on HyICE results and SGP-ExINP results will be discussed in details in future papers. Can reduce the redundancy.*

**Answer:** Yes, we agree. We removed respective statements for the HyICE campaign because not of relevance here (e.g. line 377, 474), but kept it for the DOE SGP campaign section. We suggest reformulating the sentence in line 320 to "Ongoing activities for improving the operation and data analysis tools for PINE also focus on developing an automated procedure for setting this threshold."

---

## Author Comment (AC3) · 10 Dec 2020

**Author comment in response to the comments provided by Referee #3**

We thank Paul DeMott (referee #3) for his effort in reading and commenting our manuscript. In the following, we repeat the referee's comments (italics), give point-by-point answers, and report manuscript revisions based on the referee's comments and our answers. Respective reference will be given to the line numbers of manuscript version 1.

[Figure]

**Referee comment:** *General Comments: As a scientist in this research area who both continues strong collaborations with some of the group represented on this paper and has promoted use of continuous flow diffusion chambers (CFDCs) for ice nucleation measurements over many years, I figure it is important to self-identify in this review. In this manuscript, the PINE instruments are introduced, appearing to represent a great new tool for the community, and with promise for meeting certain INP monitoring and experimental study needs. The new results for unhindered immersion freezing are very encouraging, admirable in being achieved over a relatively short period of development. From the standpoint of a publication documenting a new method, there were a few things missing for me as a very interested reader. Hence, I list a number of specific comments/questions below, driven by my desire to understand the instrument clearly. In short summary, 1) there was not a full description of principles and device(s) in order to understand what challenges may be met in applying the method for the range of ice nucleation studies inferred to be possible (deposition and immersion freezing to $-60\,°C$ (i.e., only immersion freezing is thus far discussed to the point of homogeneous freezing conditions); 2) uncertainties were given relatively limited discussion (especially at the limit of detection); 3) there was no discussion of consistency of results with physical expectations that might be revealed from, for example, microphysical modeling considerations; 4) relatedly there seemed more cursory consideration given to defining the relevant temperature associated with a measurement (I did follow the arguments, although the confirmation was mostly by comparing to AIDA), the role of growth time and sedimentation if any, clear separation of water and ice given that the latter occurs usually a few orders of magnitude lower than the water drop concentrations; and 5) finally, the introduction of field data and field instrument was rather abbreviated considering the nature/nuances of that application and considerations that will impact operation across the stated T and supersaturation range of the device in the presence of varying atmospheric conditions and full aerosol distributions. The field data only serve the purpose of demonstrating a range of data collected during automated operation for a period, as there is no other discussion of the data provided. I expect that some of the*

[Figure]

*lack of clarity that I sensed will be resolved in review here. I understand, of course, that full information on any new device is often revealed over some time, often in a number of separate publications. This is clearly underway as indicated by a paper in preparation (and other intercomparison studies I am aware of), but it suggests then that some of the statements herein may require a few caveats because supporting data are not fully shown. Hence, I might even suggest consideration of a title change to include something like "An introduction . . . " or "A first evaluation . . . " or "First description and results from . . . " or something to that effect. That is not an adamant request, simply a suggestion. The paper is otherwise well-written and an anticipated and welcome addition to the literature.*

**Answer to General Comments:**

We appreciate the comprehensive and meaningful comments from Paul DeMott, which we think not only contribute to improve the current manuscript, but are also very valuable and helpful for the further development of the new PINE instrument. We hope that our answers as well as the related revision of the manuscript will result in a sufficiently comprehensive first description of this new instrument which is clearly understandable to the interested reader.

We would also like to mention here that we purposely refrained from including everything we possibly could in this first paper, and instead address key aspects such as the operation in the field in separate papers where we can go into more detail. This first paper is intended to introduce this new instrument and to explain how it works, what it measures and how accurate the measurements are. The further development of the instrument itself and the data analysis systems are subject of ongoing activities, as the referee also noted, and will be discussed in more depth in upcoming publications, including more thorough uncertainty analysis and quality assurance measures.

Concerning general comment (1): Meanwhile, several measurements and test runs with PINE have been conducted in the cirrus cloud temperature regime. Thus, we

indeed know that PINE is capable of measuring INPs of relevance for cirrus cloud formation, but we admit that this is not the subject in this paper (our initial focus is very much on mixed-phase clouds), and we will therefore modify the manuscript accordingly (see answers to specific comments below).

Concerning general comment (3): We think the referee is mainly referring here to the proof-of-concept runs of PINE for homogeneous freezing of poor water droplets and the comparison to AIDA results. We agree that freezing of water droplets around $-35\,°C$ can also be expected from freezing rates derived from classical nucleation theory or reported in the literature. We therefore suggest to re-phrase lines 365-368 as follows:

"...and as such allowed the intercomparison of temperature-dependent freezing rates or INP concentrations. Homogeneous freezing of supercooled water droplets is known from classical nucleation theory and from literature results (Pruppacher and Klett, 2010; Koop and Murray, 2016) to occur at temperatures between about $-35\,°C$ and $-37\,°C$. Figure 7 shows the freezing of water droplets to be measured with PINE-1A in the expected temperature range. As in the experiments ...".

Setting up a microphysical model of the processes occurring within the chamber would be an interesting exercise and may become necessary in the future. However, for the study of INP relevant for mixed-phase clouds, our focus at the moment, we do not need a detailed microphysical model. The fact we create a liquid cloud defines saturation, we measure temperature directly and ice crystals are readily detected. Hence, we do not need a model to access any of the pertinent parameters. With or without a model we would place a great deal of emphasis on the comparison with AIDA, hence we focus on this comparison rather than theory.

Concerning general comment (2, 4): The referee's comments on the limited discussion of uncertainties and the comparison of PINE with AIDA refer to the more general question, whether it is possible to accurately and completely identify and specify all uncertainties of an instrument like PINE or a CFDC (Continuous Flow Diffusion Chamber), or whether a calibration to some standard or a direct comparison to a reference instrument is needed. We think the known sources of uncertainty of PINE are well mentioned and discussed in the manuscript. Up to this stage of development, we used the AIDA cloud chamber as a reference, and we think this is justified from previous intercomparison activities and results. This is our natural first order approach for an uncertainty estimate of PINE INP measurements, further systematic experiments and test are needed to quantify specific systematic uncertainties related e.g. to sampling efficiencies for aerosol and ice particles or to ice growth and size range issues. Concerning the nucleation temperature discussion, we agree that our argumentation is somehow "cursory", or to say it in other words, PINE is not capable of controlling the ice nucleation temperature as accurate as e.g. a CFDC is. This is one of the limitations of PINE, but when measuring immersion INP concentrations over a wide temperature range from about $-10\,^{\circ}$C to $-35\,^{\circ}$C, an overall temperature uncertainty of $\pm 1\,^{\circ}$C according to current conservative estimates may be sufficient to quantify the temperature spectrum of INP concentrations.

Another important parameter for long-term observations, the PINE instrument is also developed for, is the precision for repeated measurements at the same sampling and operating conditions. Figure 1 shows a recent measurement with the new commercial PINE-04-01 when sampling a mixed aerosol (ammonium sulfate and natural dust) for more than 8 hours from the AIDA chamber. This figure well demonstrates the run-by-run stability and repeatability of PINE measurements. In this experiment we did not expect a constant but a steadily decreasing INP concentration (panel a), according to the steady decrease of the aerosol concentration (panel b) according to aerosol loss processes to the chamber walls. The ice-active particle number fraction (panel c) remained constant with a mean value of $1.8 \times 10^{-4}$ and a standard deviation of $2.1 \times 10^{-5}$, or a relative uncertainty of about $12\%$, which demonstrates the precision of PINE INP measurements under these conditions. During this operation, an average number $N_{ice}$ of about $90$ ice crystals was measured during one run. Therefore, the relative uncertainty from counting statistics can be calculated as $\sqrt{N_{ice}}/N_{ice} = 10.5\%$,

which is close to the observed standard deviation.

We will include this figure and the respective text to the appendix C (new Figure A6). We also suggest to modify and extend the last sentence of section 5 (lines 407 to 410):

"This also underlines the assumption, that the ice formation in PINE is mainly controlled by the coldest temperature in the bottom part of the chamber and that the number concentration of ice crystals, and by that the number concentration of ice-active aerosol particles in laboratory experiments and of INPs during field measurements can be calculated with Eqs. 5 and 6 within the above given uncertainty estimates for the number concentration and the nucleation temperature. These estimates are justified by the comparison of PINE with AIDA results. Further systematic uncertainties like the loss of large ice crystals between the PINE cloud chamber and OPC, size range overlap of small ice crystals with large aerosol particles not activated to droplets, or the sampling efficiency of large aerosol particles into the cloud chamber may have to be considered for calculating the overall accuracy of INP measurements.

A more comprehensive uncertainty assessment for PINE may result from recent inter-comparison studies with other methods and instruments and ongoing long-term operation in field campaigns. For long-term measurements, another important parameter is the precision for repeated measurements at the same sampling and operating conditions. In a recent test experiment at the AIDA cloud chamber, the new commercial PINE-04-01instrument sampled a mixed aerosol (ammonium sulfate and natural dust for more than 8 hours from the AIDA chamber (Figure A6). During this experiment, a mean ice-active particle number fraction of $1.8 \times 10^{-4}$ was measured with a standard deviation of $2.1 \times 10^{-5}$, which corresponds to a relative uncertainty of about $12\%$. During this operation, an average number Nice of about $90$ ice crystals was measured during one run. Therefore, the relative uncertainty from counting statistics can be calculated as $\sqrt{N_{ice}}/N_{ice} = 10.5\%$, which is close to the relative standard deviation of the run by run data from the mean value. For measurements with a much lower number of ice crystals detected in one run or a consecutive number of runs, the measurements

uncertainty from counting statistics can be much larger. Next versions of the PINE analysis software tools will also include uncertainty analysis for low counting cases close to the PINE detection limit."

We also noted that there is an error in line 401. The temperature uncertainty during an AIDA cloud expansion chamber is noted as $\pm 1\,^{\circ}$C. This should be corrected to $\pm 0.3\,^{\circ}$C.

Concerning general comment (5): The description of the field instrument was brief because it's operating principles are identical. Also the discussion of the data was brief because we are planning a succession of papers focused on the field data (as the referee notes).

Concerning the title: We feel our existing title is accurate, and the phrase "a new" implies that this is the first description.

Further suggested changes to the manuscript will be included below along with our answers to the specific referee comments.

**Specific Comments**

**Introduction:**

**Referee comment:**

*1) The introduction was comprehensive, perhaps overly so for a paper describing a new instrument. It was long, and not so much related to the development itself other than attempting to meet motivations.*

**Answer:**

Referee 1 had a similar comment. See answers given there. Further revision is mentioned below in response to more specific comments.

**Referee comment:**

*2) Lines 87-89: It seems clear that both low and high time resolution are desirable*

*things for different scenarios. High time resolution is arguably not useful if one is attempting to document the most special INPs, the ones that even the PINE instrument may struggle to measure in all but INP-rich environments such as shown later in this paper. I see higher time resolution measurements as highly useful, but not sufficient, unless their resolution can match higher volume collections. Some of the studies referenced to preface this statement were made with instruments capable of even higher time resolution than the PINE, but the issue I am speaking of is resolving low INP concentrations in some environments. Those other methods have been developed for automation as well, a point I will raise next.*

**Answer:**

We agree the important point are missing here, and suggest to modify lines 87-89 as follows:

"Depending on the specific campaign goals and objectives, different instruments and methods were used like CFDCs with higher time resolution to e.g. characterize changing air masses (e.g. Boose et al., 2016a; Lacher et al., 2018), or aerosol filter based offline methods to achieve high sensitivity for characterizing INPs at higher temperatures or in clean environments (e.g. Wex et al., 2019), or a combination of both (e.g. Welti et al., 2018). What is missing so far are long-term monitoring of INPs with high time resolution and over a wide temperature range."

**Referee comment:**

*3) Lines 97-100: I would say to be fair that these statements need modification or qualification for other studies in the recent literature. I think that the continuous flow chamber developments reflected in Bi et al. (2019) and Brunner et al. (2020) meet the criteria of operating more than "periodically" and of saving "intensive man-power and time for operation or offline analysis." Such developments are advancing rapidly across the discipline. Those instruments also appear to be capable of higher time resolution than is demonstrated for the PINE instrument in this paper. It is also implied*

*here that time resolution of minutes will somehow solve the INP size and chemistry resolution issue, although due to statistics (sample volume and particle numbers) it is hard to imagine this as yet being achievable for single INPs except in high loading situations. Rather, this would occur by correlation to independent high resolution composition measurements for all aerosols, which sometimes does not work for comparing directly to specialized INPs. Hence, I see high resolution INP capabilities as one piece needed in the course of a full development.*

**Answer:**

Thanks for this very valuable comment, and thanks for referring to Bi et al. (2019) and Brunner and Kanji (2020). In fact, we have not cited Bi et al. (2019), and this error has been rectified. Brunner and Kanji (2020) was available as a discussion paper when writing, and it will also be cited in the revised paper. We also agree that other statements in this section are somehow imbalanced and suggest to re-phrase lines 94 to 100 as follows:

"Most of the INP methods showed reasonable agreement with each other, but many of them are time and operator intensive. A general feature is, that offline methods based on aerosol filter samples have poor time resolution depending on required aerosol sampling time of hours to days, and require intensive man-power and time for both operation and offline analysis. Most online instruments can only be operated periodically, and also require operator time during the measurements, but can be operated for INP measurements at higher time resolution in particular at low temperature or in polluted environments where concentrations are higher. Only recently, newly developed INP instruments with a higher degree of automation became available (Bi et al., 2019; Brunner and Kanji, 2020). The automated CFDC instrument used by Bi et al. (2019) performed INP measurements during a period of one month in 2018 at temperatures between $-20\,°C$ and $-30\,°C$. The CFDC instrument called HINC-Auto (Horizontal Ice Nucleation Chamber) used by Brunner and Kanji (2020) autonomously measured immersion freezing INP for $90$ consecutive days, but only at one temperature of $-30\,°C$.

A combination of both, high time resolution and wide temperature range for long-term INP measurements, together with a comprehensive set of high resolution aerosol analytics, would challenge the comparison to potential driving factors for atmospheric ice nucleation."

**Referee comment:**

*4) Line 103: The stated temperature range is what the instrument is designed for, but no exploration of capabilities to make useful measurements to as low as $-60\,°C$ are given in this manuscript. It appears as a potential capability, only in that the temperature can be achieved in PINE-c. One can imagine that challenges in operating and interpreting data to that lower limit could be significant (e.g., control on final T and RH of expansion, low water vapor pressure and slow ice crystal growth rates), and not simply depend on the capacity of the cooling system (line 121). I suggest to stick to what is demonstrated in this paper, as far as confirmed operational capabilities and to clearly identify capabilities that remain to be defined.*

**Answer:**

We agree and suggest to modify lines 103/104 to "...in a wide temperature range. This paper demonstrates the instrument's ability to measure in the mixed-phase cloud temperature regime from $-10\,°C$ to $-40\,°C$. PINE is also able to measure ice nucleation at cirrus cloud temperatures to about $-65\,°C$, which is the topic of ongoing studies." (see also answer to comments from referee #1).

**Basic principles and milestones of the PINE development:**

**Referee comment:**

*1) Line 121 repeats the assertion that likely requires "potential" as a caveat. No low temperature data are shown excepting the homogeneous freezing onset for grown droplets.*

**Answer:**

We suggest to remove "and thereby the temperature range of ice formation and INP detection"

**PINE instrument setup:**

**Referee comment:**

*1) Line 169 or thereafter: Have particle losses been characterized through the nafion dryer system? For that matter, I realize in reviewing these comments that particle transfer versus size into the PINE systems has not been discussed.*

**Answer:**

We characterized the particle loss through the dryers at the NAUA aerosol chamber, using a natural dust sample from Marocco. An APS was used to measure the aerosol size distribution before and after the dryers, which were mounted in a way to represent their orientation at PINE, with vertical orientation and the sampled air flowing in upward direction, then bending by 180 degrees for straight downward flow into the PINE cloud chamber. Figure 2 shows both the aerosol particle size distribution with and without the dryers, demonstrating that the loss of particles in the size range of up to about $2\,\mu m$ is minor. Only particles larger than approximately $4\,\mu m$ experience a major loss of more than $50\%$. As such we have confidence that a large fraction of atmospherically relevant particles will enter PINE. In setups were the sample flow can be taken in strictly vertical downward orientation, the particle loss can be expected to be much smaller. More systematic experiments of this kind will be performed in the future, where the particle loss in the dryers and in the PINE chamber will be characterized.

We suggest adding the following paragraph at line 180:

"In the commercial version, the standard location of the dryers is next to the cloud chamber with vertical orientation, so that the sampled air flows in upward direction through the dyers, then passes a $90°$ bend, a horizontal distance of $50\,cm$ and another $90°$ bend to then flow downward into the PINE cloud chamber. The aerosol particle

loss for this setup was measured to be less than $20\%$ for particles smaller than $2\,\mu\mathrm{m}$ diameter. It decreased to about $50\%$ for particles with an aerodynamic diameter of about $4\,\mu\mathrm{m}$. The dryers can also be mounted above the PINE chamber for a strictly vertical sample flow, for which a further reduced particle loss can be expected. More systematic sampling efficiency measurements for different configurations and operations will be performed in the future."

**Referee comment:**

*2) Lines 172-173: Perhaps this is irrelevant since an aircraft system is not yet described, but I wondered about the use of the nafion system on aircraft where the pressure drop will be limited at higher altitudes. Will the system work over the needed ranges in this scenario?*

**Answer:**

We know from first estimates and test series that a dryer will not be needed when sampling dry air in the middle/upper troposphere. We may even need a humidifier instead, depending on air temperatures and relative humidity. Nevertheless, the nafion system can still be operated at reduced absolute pressure, but what makes it inefficient for application in the free troposphere is not the reduced pressure drop over the membrane (you could even think of operating the dryer with dry synthetic air) but the drying efficiency of the membrane itself which seems to be limited to an absolute frost point temperature of about $-20\,^\circ\mathrm{C}$.

**Referee comment:**

*3) Lines 177-180: I am curious about the later tests shown for background, simply because I did not understand the implications of no background particles found after five runs. Why does it take five runs to decrease, and does it mean that any background is then absent from thence forward in time? Have you explored this systematically, and/or after hours of operation? My personal understanding from an overlapping study*

*in time with the one at the SGP site, is that the dewpoint was $-10\,°C$ in that case, and that background counts at some level were always detected, if minimal. Hence, the basic question is if it is understood what dewpoint is sufficient for frost-free operation at any given T?*

**Answer:**

We have to distinguish here between two sorts of background, one coming from frost build-up on the cold chamber walls during longer time operation at temperatures lower than the frost point temperature of sampled air, and the other coming from large aerosol particles or liquid droplets overlapping in size with ice crystal detection. Here we only argue about the absence of the background from frost artifacts when operating PINE with filtered, particle free air. It takes up to five runs to completely flush the chamber with filtered air and to achieve particle free conditions. Only then we can be sure that any remaining ice counts would come from frost at the walls. In longer term operations of PINE, we do such frost background checks not only once but usually repeat them every day, so we already tested the long term behavior for frost artefacts. To make this more clear we suggest to rephrase line 178 to

"... resulting in zero particle counts in the detection range for ice crystals after about 5 consecutive runs ... ', and to add in line 180 "Such frost background tests are usually repeated once every day in long term operation of PINE."

**Referee comment:**

*4) Lines 183-184: It was not clear to me what actually constitutes the cooling system? Is it a plenum around the chamber and this is fed by the large chiller reservoir?*

**Answer:**

Good point. The PINE-1A cloud chamber is actually cooled by circulating ethanol from the bath chiller through special thermo-conductive plastic tubes wounded around the cloud chamber. To make this clear, we suggest adding the following sentence in line

185:

"This is achieved by circulating the chilled ethanol from the bath chiller through thermo-conductive EPDM (ethylene propylene diene monomer rubber) tubes wounded around the chamber."

We also suggest removing the word "precisely" in line 183.

**Referee comment:**

*5) Line 191: I expected that the minimum air temperature achieved would be colder than the minimum cooling temperature? Why are they the same?*

**Answer:**

Thanks for this comment. The minimum gas temperature reached at a wall temperature of $-33\,°C$ is about $-40\,°C$. We will correct this error. Related to this, we noted that a wrong lower limit for the PINE measurement range is given in line 7 of the abstract. This will be changed from $-38\,°C$ to $-40\,°C$.

**Referee comment:**

*6) Line 195: Can you explain the Stirling cooler method of cooling the wall of PINE-c for those of us unfamiliar with the exact cooling mechanism? E.g., fluid versus expansion cooling or whatever it is. The details on cooling systems in general does not match the later attention to detail of the OPCs.*

**Answer:**

We suggest to add the following text at line xy of the manuscript:

"A dual opposed pistons compressor driven by linear motors with moving magnet flexure bearing design drives a Stirling-type pulse tube. As a consequence, there is only little vibration introduced to the cloud chamber in direct thermal contact to the pulse tube. The compressor of the cryocooler is force-flow air-cooled. Therefore, no cooling

liquids are required and the cooling system is maintenance-free."

We also suggest to add the following reference:

D.L. Johnson, I.M. McKinley, J.I. Rodriguez, H. Tseng, and B.A. Carroll, Characterization testing of the Thales LPT9310 pulse tube cooler, in Cryocoolers 18 (S.D. Miller and R.G. Ross, Jr., eds.), pp. 125–133, Plenum Press, 2014.

**Referee comment:**

*7) Line 198-200: Again, the cooling is understood, but the utility for performing low temperature ice nucleation experiments, especially where this will presumably involve more special control over the expansion conditions to meet some final peak relative humidity, is not yet discussed or demonstrated herein.*

**Answer:**

OK, but we still would like to mention here the technical capabilities for future work with PINE and therefore suggest to modify lines 198-200 as follows: "PINE-c can also be cooled to a lower wall temperature of $-60\,°C$ and can therefore be operated at cirrus cloud temperatures in upcoming studies."

**Referee comment:**

*8) Line 222: Does this more limited volume used to define ODV explain the higher value of lowest detection limit concentration listed in Table? Perhaps worth noting here, since it only comes up again at the end of section 4.*

**Answer:**

Yes, this is indeed the reason for the different detection limits. At the end of the same paragraph we already mentioned the detection limit to depend on the volume flow through the OPC. Therefore, we do not see a need for change or extension here.

**PINE operating principle:**

**Referee comment:**

*1) Line 250: To this point, the definition of ice crystals versus drops has not been made. Perhaps add a short note about this, ". . . as discussed later in this section"? Otherwise, this raised a number of questions immediately.*

**Answer:**

Good point. We suggest to change line 249 to ". . . are then activated to form liquid cloud droplets and/or ice crystals, depending on . . . ", and to re-phrase the sentence in line 250: "Both droplets and ice crystals are measured with an OPC downstream of the chamber. Ice crystals are distinguished from droplets by their larger optical size, as discussed later in this section."

**Referee comment:**

*2) Lines 269-270: Figure 3 is an important figure, and it raises a number of questions that were mostly answered in time over this section. However, I will list a number of them here. Immediately I wondered why the lowest temperature measured was used. As an aside, this point (lowest T used) should also be stated in the figure caption, for clarity. What differences are seen in these temperatures, and what uncertainty does this create? Are concentrations referenced to the entire integrated time interval and volume of expansion (and will this be the case also for the PINE-c), and do they represent the lowest temperature achieved (e.g., there is a $4\,°C$ cooling shown in the figure over the time of the expansion)? Hence, is it one measurement or many, and how are the sub-intervals defined? A range of apparent ice crystal sizes are shown in Fig. 3, up to 100 microns. Are these ice sizes consistent with expectations of grown sizes for the conditions and growth times? The PINE chamber is quite small compared to the AIDA chamber where volumetric concentrations are assessed in situ. Is there sedimentation that could impact inferred concentrations and their reference temperature for the smaller geometry of the PINE? Have any such calculations been made at this time, or are they planned?*

**Answer:**

We agree that this is an important figure, but more in the sense of explaining the basic measurement principle of PINE and the three different modes of what we call a run. The temperature chosen for this plot is only of minor importance here, but we agree to add this information to caption of Figure 3. The relation of increasing number of ice with decreasing temperatures in the course of an expansion is discussed in lines 330 to 345. In runs with a larger number of ice crystals, one may obtain a number of INP data points in certain temperature subintervals. We analyzed the PINE data in this way in a number of field and laboratory based operations, and may come back to this approach on future publications. For now, we decided to report the cumulative number of INPs that corresponds to the minimum nucleation temperature in a PINE run. The analysis program sums up all ice crystals detected during one run, and calculates the number density by dividing this number by the total volume that passed the OPC. According to the ice nucleation active surface site density concept, this cumulative number of ice is well defined and independent of the start temperature for droplet and ice formation during the expansion mode.

Concerning the temperature uncertainty see our answer to the general comments above. The reason and justification for using the lowest temperature measured as the "nucleation temperature" is mentioned in lines 343 to 345, and also the results of comparing PINE with AIDA results (see Figures 7 to 11).

Concerning the ice crystal size, please note that these are optical sizes. We know from experience with welas measurements at AIDA, and from scattering phase function calculations, that the sideward scattering geometry of both the welas and fidas sensors detect a-spherical particles with a much larger scattering intensity than spherical particles of the same volume and refractive index. Järvinen et al. (2014) determined an average oversizing factor of 2.2 for a welas sensor. For individual ice crystals, this factor can be much larger depending on their size, shape and orientation in the OPC detection volume. Therefore, the geometric size of ice crystals is much smaller than

shown in Figure 3. We suggest to add the following text at the end of line 306:

"The use of a simple size threshold to distinguish between ice crystals and droplets is supported by the fact that the sideward scattering geometry of both the welas and fidas sensors detect a-spherical particles with a much larger scattering intensity than spherical particles of the same volume and refractive index. Järvinen et al. (2014) determined an average oversizing factor of 2.2 for the welas sensor. For individual ice crystals, this factor can be much larger depending on their size, shape and orientation in the OPC detection volume."

In this paper, we suggest to stay with demonstrating and documenting the quality and accuracy of PINE measurements and data analysis procedures by comparison to AIDA results (see also our answers to the general comments above).

Additional reference:

E. Järvinen, P. Vochezer, O. Möhler, and M. Schnaiter, "Laboratory study of microphysical and scattering properties of corona-producing cirrus clouds," Appl. Opt. 53, 7566-7575 (2014).

**Referee comment:**

*3) Lines 273-275: Regarding the starting vapor saturation ratio for expansion, you assumed this or you set that partial pressure based on a room temperature RH measurement? Why would it be ice saturated if there is no ice on the walls? Or is it close enough as determined on some other basis? This would seem important for future use toward other measurements than immersion freezing.*

**Answer:**

We did not say that there will be no ice at all at the wall. We only stated that no frost fragments are observed even after any deposits have eventually accumulated over longer operation periods while sampling slightly ice supersaturated air. When the frost point temperature of air added to the cloud chamber is higher than the wall temperature

we assume that the excess water vapor still deposits to the wall so that ice saturated conditions are reached or at least approached at the beginning of the expansion mode. Part of this wall ice deposit may be removed again when refilling the cloud chamber which causes an adiabatic warming of the gas inside the cloud chamber. But the referee is raising an important point here which we will have in mind for further test series, in particular at lower temperatures of PINE operation.

**Referee comment:**

*4) Line 288: Here an important distinction may arise, but perhaps the authors can correct any misconception I have. While described as purely immersion freezing, the temperature is already cold at the point of expansion, and so does the measurement also not integrate some proportion of INPs from any/all INP mechanisms, other than contact freezing, that ensue as the air rapidly cools and ultimately exits the chamber through the OPC? That is, somewhat similar to CFDCs when they are operated for bringing air to a final RH that is well above water saturation?*

**Answer:**

Yes, the referee is right, other modes of nucleation are possible. In the case of very high aerosol concentrations in an AIDA cloud expansion experiment a clear development of a supercooled droplet cloud does not necessarily occur in the course of an expansion. We did observe this case in recent laboratory tests and calibration runs with high INP number concentrations. In all field operations so far we always saw a clear development of a supercooled droplet cloud and only a minor number fraction of ice crystals, as in the case mentioned in line 288. In this case we believe that most, if not all ice observed can only be formed by immersion freezing INPs.

**Referee comment:**

*5) Line 317-318: Concerning addressing the size threshold setting for ice crystals, I struggled a bit to reconcile Figures A5 and 5. In A5, the scale is frequency, and it spans*

*about three orders of magnitude out to 10 microns. Is there an issue in the fact that if cloud droplet concentrations range up to 1000 per cubic centimeter, and activated INP concentrations could range down to 1 per liter, then assessment of cloud drop frequency would have to be made over a greatly extended period of time to capture the tail of the distribution? Or is it simply the case that repeated experiments like the one in Fig. A5 never indicated a drop even in the size range greater than 10 microns? It might help to add the time and/or volume of assessment represented in Fig. A5. Clearly, Fig. 5 shows particle numbers appearing in these larger size ranges at 4-5 orders of magnitude below cloud droplet concentrations at least. This is an issue that perhaps deserved more attention in the paper, but if I understand, sensitivities of the ice cut size threshold will be more extensively covered in Adams et al. It would be good to add a reference for that paper, if it is now in submission.*

**Answer:**

Figure A5 is just one example of a droplet size distribution without the presence of INPs. Of importance is here the sharp edge of the size distribution, which we also observed in many other cases. Given that the expansion is rapid, there is little opportunity for some droplets to grow more than others and a tail of the droplet size distribution towards larger diameters was not observed so far. This is also expected given the diameter growth rate of a spherical droplet in the continuum regime to be inversely proportional to its diameter. Systematic uncertainties related to the size threshold may more result from a potential overlap of the ice crystal size distribution with the droplet size distribution, less to a tail of the droplet size distribution towards ice crystal sizes. We have selected a safe threshold size to be sure we never catch the high size end of the droplet distribution, at the expense of eventually undercounting the ice crystal concentration.

This will indeed be investigated in more detail in upcoming publications. The one by Adams et al., however, is not yet submitted. We will therefore remove reference to this paper or replace by statements like "will be discussed in more detail in upcoming

publications".

**Referee comment:**

*6) Lines 348-349 and lines 359-360: Note that the first statement repeats from earlier in the manuscript. One example is provided in Fig. A3. Perhaps repeating myself also, is this the very start of operations, or a period during the midst of operations? Why does it take 5 cycles at all, and does the background then stay that low in all cases? What does this depend on? The question arises again in the later sentence where long time operating detection limits are listed. Do not these very low detection limits listed for long operation imply the need for validating backgrounds being below such levels over such long times?*

**Answer:**

As already stated above, it takes several cycles with filtered air to remove all or at least most of the aerosol particles and by that also the INPs from the cloud chamber and by that then prove that frost background is indeed zero. Such background operations are then repeated at least once a day to check for frost to accumulate or not. When not frost is accumulated, then we consider the chamber walls to stay free of accumulated frost formation and by that free or background frost artefacts. We agree to the referee that such background test have to be done also over longer operation times and already started to do so.

**Laboratory tests of the prototype version PINE-1A:**

**Referee comment:**

*1) Lines 374-375: In Fig. 6, there looks to be up to 1C temperature uncertainty in defining the lowest temperature attributed to ice nucleation. Since T is not spatially uniform in the chamber, do you anticipate a bias in sampling only part of the flow as in PINE-1A versus all of the flow in PINE-c? Also, Figure 6 and its caption could use a little attention to description. At present the data are described as "all single ice*

*crystals measured." Should it say something like "Data points indicate all single ice
crystal event temperatures . . . "*

**Answer:**

In a number of test runs with PINE-1A using either welas or fidas we did not observe
a difference in the freezing temperature measured for pure water droplets. Such a
difference or bias is also not expected because it is the same portion of air pumped
through the OPC, just a different fraction analyzed. We agree to modify the caption of
Figure 6 to

"The data points show event temperatures of all ice crystals measured with PINE-1A
. . . in Figs. 4 and 5. The events are plotted as a function of the relative run time they
were detected and the gas temperatures measured at the same time with three sensors
. . ."

**Referee comment:**

*2) Line 391: Just a note that there seems an inconsistency between the statement of a
minimum pressure reduction every 5th cycle versus what is shown in Fig. 8 (and stated
in that caption). It looks like 4 cycles. It is 5 cycles in Fig. 10.*

**Answer:**

Thanks for noting this. It is even every third run in the example shown in Fig.8. We
correct the text body and the figure caption for this.

**Referee comment:**

*3) Lines 401 to end of section: The basic agreement shown between AIDA and PINE in
Figures 7, 9 and 11 (over a more limited range) is excellent. I again wonder here about
the percentage uncertainties being constant over the entire dynamic range of ice con-
centrations. For example, at the LOD, the true uncertainties must be larger, no? That
statistical uncertainty does not appear to be captured in defining uncertainties based*

*on the OPC ODV alone. I guess I expected based on statistical count considerations that the uncertainties should be larger for lower INP concentrations. Additionally, given that ice concentrations are integrated over the range of temperatures present through-out the volume, and if some of the crystals grow in that time to 50-100 microns (would be good to state the typical mode size), does sedimentation assuredly not impact/skew the results attributed to one temperature? There could be differences as to how this is measured temporally in situ in AIDA versus drawing the entire tank flow from the PINE, and there is some room for not discerning that in the comparisons shown. Neverthe-less, a minor point overall.*

**Answer:**

See our answers given to the general comment above, and the new Figure A6.

**Field measurements with PINE-c:**

**Referee comment:**

*1) Lines 416-417: With an expansion mode time of $60$ to $90$ s, a question arises as to the applicability of the discussion of temperature attribution and method for calculating INP concentrations with the PINE-c versus PINE-1A. Were they exactly the same (lowest T used, etc) for these presented analyses?*

**Answer:**

Yes, for both instruments, we analyzed and plotted the INP number concentrations for the lowest temperature reading, and used the same equations as discussed in the manuscript.

**Referee comment:**

*2) Fig. 12: This is a nice compilation of results, if leaving a lot of room for discussion of their meaning still (i.e., variability of 2 orders of magnitude temporally at any T,) and raising all of the questions listed in the last sentence of this section. It is a minor*

*concern for showing them in this manner, simply as a demonstration that the data were collected more or less autonomously over this period (maintenance or other attention needed were not discussed). Let me ask one thing though. The flattening of the INP concentrations toward the higher temperature limit of detection is interesting, but raises a question regarding the confidence in these results. The uncertainties are based on relative standard errors. The percentage errors are quite small and I wonder how these can be the same at the LOD as they are at any other conditions. This is the same question raised for PINE-1A.*

**Answer:**

We intend to introduce these compiled quasi-raw PINE data in a snapshot single figure to demonstrate the PINE's capability for fairly long-term continuous operation in a simple manner. Nevertheless, we certainly understand the reviewer's concerns - there is ample room for further discussion on many details. To mitigate the reviewer's and reader's misgivings, we have revised our sentence in L425-427 to address remaining items to be investigated in the future (please see our response below). The nice uncertainties in this figure are based on relative standard errors of time-averaged data, which appear to be small - equivalent to or smaller than the "systematic" error of OPC ($\pm 20\%$). We have started to run detailed statistical error analysis with an inclusion of estimated backgrounds for a subset of our PINE-c field data, and confirmed that the nice uncertainty near the LOD at relatively high temperatures propagates and becomes apparent, as a relative importance of background contribution becomes prominent in such temperatures. Some of the authors of this paper will carefully characterize the data in this region and address our findings along with other detailed topics (i.e., L425-427) in our future paper. We suggest to add the following sentence at the end of the Figure 12 caption: "Statistical errors from low counting signals are not considered here and will be the subject of further analysis."

**Referee comment:**

[Figure]

*3) Lines 423-424: I am not sure what is meant by "warranted" here. Possible? Also, can the point regarding the dewpoint temperature be clarified? Dewpoint is not controlled somehow? It would be much higher in summer and much lower in winter. How might this affect the operational range, background etc, or does this remain to be investigated?*

**Answer:**

Yes, "possible" is a better and more appropriate word choice here. Furthermore, thanks to the referee's comment, we noted that we should refer here to the dew point temperature of the sample air after passing the dryer, and not to the dew point of the ambient air. We therefore suggest to rephrase lines 423-424 as follows:

"This temperature range represents the PINE-c condition, where ice nucleation through immersion freezing was possible below the frost point temperature of the sample air, which passes the membrane diffusion dryers operated at maximum drying efficiency. For measurements at higher temperature, the drying efficiency has to be reduced, in order to increase the dew point of the sampled air and to exceed water saturation during the expansion mode at higher temperature. Next versions of the PINE control program will include this option for operation at higher temperature."

**Referee comment:**

*4) Lines 425-427: What exactly is meant by deconvolution of nucleation modes? Meaning different operation of the PINE than discussed in this paper, which is immersion freezing? Or meaning resolving what I mentioned earlier in this review, the temporal evaluation of data during single expansions? This is a point that should be clarified, as it is important to state which potential aspects of PINE measurement capabilities are demonstrated in this first publication and which remain.*

**Answer:**

It was meant for separating/estimating ice crystals formed through immersion freezing

from other ice nucleation paths, however not just from PINE measurement but in combination with all other instruments data and measurements. PINE alone will only be able to measure immersion freezing in the temperature range of interest in this paper. We admit the sentence in lines 425-427 is confusing. For clarity, we suggest to revise the sentence as:

"Any further scientific discussions regarding PINE-c operations and observations, in combination with other INP and aerosol measurements during the ExINP-SGP campaign, are beyond the scope of our current study, and will be followed up in future publications."

**Other editorial comments:**

*Line 96: typo - based*

Corrected

*Line 266: Suggest "one of" after "An example of . . . "*

Added

*Line 282: "so" not needed before "calculated"*

Removed

*Line 287: Suggest "than" for "as"*

Changed

*Figure 3 caption: Suggest to add "Calculated" at start of sentence starting "Liquid water . . . "*

Changed

References:

Bi, K., G. R. McMeeking, D. Ding, E. J. T. Levin, P. J. DeMott, D. Zhao, F. Wang, Q.

Liu, P. Tian, X. Ma, Y. Chen, M. Huang, H. Zhang, T. Gordon, and P. Chen, 2019: Measurements of ice nucleating particles in Beijing, China. Journal of Geophysical Research: Atmospheres, 124, 8065–8075. https:// doi.org/10.1029/2019JD030609

Brunner, C. and Kanji, Z. A.: Continuous online-monitoring of Ice Nucleating Particles: development of the automated Horizontal Ice Nucleation Chamber (HINC-Auto), Atmos. Meas. Tech. Discuss., https://doi.org/10.5194/amt-2020-306, in review, 2020.
* * *
[Figure]

[Figure]

**Fig. 1.** Figure 1: Time series of INP concentration (a), total aerosol particle number concentration (b), and the ice-active particle fraction (c).

**Fig. 2.** Aerosol particle size distribution of a natural dust sample measured directly from NAUA (black bars) and downstream of the nafion dryers (blue bars), and the percentage particle loss (red stars).

---

## Author Comment (AC4) · 11 Dec 2020

**Author comment in response to the comments provided by Referee #4**

We thank referee #4 for her/his effort in reading and commenting our manuscript. In the following, we repeat the referee's comments (italics), give point-by-point answers, and suggest manuscript revisions based on the referee's comments and our answers. Respective reference will be given to the line numbers of manuscript version 1.

**Referee comment:**

[Figure]

*The knowledge of ice nucleating particles and their impacts on clouds was restricted by the development of measurement techniques and instruments. This manuscript presents a new instrument based on expansion chamber for both laboratory studies and field observations to measure ice nucleating particles. The authors successfully demonstrate the applicability of their new instrument to be compared with AIDA and deployed in a field campaign. Different from the commonly used Continuous Flow Diffusion Chamber (CFDC), PINE is truly the first commercial instrument capable of automated long-term continuous observation, and its development provides an excellent complement to enrich measurement methods. This research aligns well with the scope of AMT. The manuscript is well written and easy to follow, thus should be acceptable for publication after considering following minor comments:*

**Referee comment:**

*P2 Line28: The abbreviation "(INPs)" should not be linked directly after ". . . atmospheric aerosol particles". Moreover, two sets of parentheses are used instead of one. For example, ". . . atmospheric aerosol particles (INPs) (Vali et al., 2015)".*

**Answer:**

Thaks for noting this. We suggest to reformulate lines 27/28 as follows:

"In the absence of homogeneous freezing, the cloud ice phase is initiated in various ways by ice nucleating particles (INPs), a very small fraction of atmospheric aerosol particles (Vali et al., 2015)."

**Referee comment:**

*P2-3: The need for promoting INP monitoring was put forward until P3 Line 63, and the new methods and instruments for INP measurements were discussed from P3 Line 90. Please consider adding more descriptions and comparisons of existing instruments, especially the most commonly used CFDC, and simplifying the context on INP.*

**Answer:**

Referees #1 and #3 have also suggested similar revisions to the introduction. We have removed some parts and in particular revised lines 84 to 100 (see our answers to referees #1 and #3). We have in particular added reference to recent developments of by Bi et al. (2019) and Brunner and Kanji (2020), and for other instruments like CFDCs we referred the reader to the paper by DeMott et al. (2018).

**Referee comment:**

*P9 Line275: A parenthesis is missed.*

**Answer:**

Yes, added.

**Referee comment:**

*P12-14: PINE-c performed the field measurements to demonstrate its capability, however, the comparisons with AIDA and performance tests were conducted by the prototype version PINE-1A. PINE-c is a further developed version with major upgrades in chamber type, cooling system, controlled temperature range, particle detector, and so on. So direct characterizations and tests of the performance of PINE-c would be helpful.*

**Answer:**

This is a good point. Unfortunately, there was time for only a few test runs of PINE-c at the AIDA (Aerosol Interaction and Dynamics in the Atmosphere) cloud chamber facility, before the instrument had to be delivered for participation in the ARM-SGP field campaign. Meanwhile, three more instruments of type PINE-c have been built, and two of them are currently extensively tested and operated at the AIDA facility (see for instance the new Figure A6 in the appendix) and also next to the PINE-1A version. A more comprehensive comparison of PINE-c with other methods and instruments will be teh subject of upcoming paublications.

**Referee comment:**

*Figure 7: Please notice the superscript of the unit, ". . . 5 l min-1".*

**Answer:**

Yes, corrected.

**References:**

Bi, K., G. R. McMeeking, D. Ding, E. J. T. Levin, P. J. DeMott, D. Zhao, F. Wang, Q. Liu, P. Tian, X. Ma, Y. Chen, M. Huang, H. Zhang, T. Gordon, and P. Chen, 2019: Measurements of ice nucleating particles in Beijing, China. Journal of Geophysical Research: Atmospheres, 124, 8065–8075. https:// doi.org/10.1029/2019JD030609

Brunner, C. and Kanji, Z. A.: Continuous online-monitoring of Ice Nucleating Particles: development of the automated Horizontal Ice Nucleation Chamber (HINC-Auto), Atmos. Meas. Tech. Discuss., https://doi.org/10.5194/amt-2020-306, in review, 2020.

DeMott, P. J., Möhler, O., Cziczo, D. J., et al.: The Fifth International Workshop on Ice Nucleation phase 2 (FIN-02): laboratory intercomparison of ice nucleation measurements, Atmos. Meas. Tech., 11, 6231–6257, https://doi.org/10.5194/amt-11-6231-2018, 2018.

---

## Author Response (AR2)

**Final author response for manuscript amt-2020-307**

Dear Hang Su,

on behalf of all co-authors, I would like to thank you again for handling our manuscript so well. Final acceptance is like another Christmas present to us. I am happy to now upload the final versions and documents.

With best regards,

Ottmar Möhler